# FLUX-Reason-6M & PRISM-Bench: A Million-Scale Text-to-Image Reasoning Dataset and Comprehensive Benchmark

**Rongyao Fang[1,4]\*,  Aldrich Yu[1]\*,  Chengqi Duan[2]\*,  Linjiang Huang[3],  Shuai Bai[4],**
**Yuxuan Cai[4],  Kun Wang[5],  Si Liu[3],  Xihui Liu[2‡],  Hongsheng Li[1‡]**

[1]CUHK MMLab    [2]HKU MMLab    [3]BUAA    [4]Alibaba    [5]Sensetime

\*Equal Contribution    ‡Corresponding Author

## ABSTRACT

The advancement of open-source text-to-image (T2I) models has been hindered by the absence of large-scale, reasoning-focused datasets and comprehensive evaluation benchmarks, resulting in a performance gap compared to leading closed-source systems. To address this challenge, We introduce **FLUX-Reason-6M** and **PRISM-Bench** (**P**recise and **R**obust **I**mage **S**ynthesis **M**easurement **Bench**mark). FLUX-Reason-6M is a massive dataset consisting of 6 million high-quality FLUX-generated images and 20 million bilingual (English and Chinese) descriptions specifically designed to teach complex reasoning. The image are organized according to six key characteristics: *Imagination*, *Entity*, *Text rendering*, *Style*, *Affection*, and *Composition*, and design explicit Generation Chain-of-Thought (**GCoT**) to provide detailed breakdowns of image generation steps. PRISM-Bench offers a novel evaluation standard with seven distinct tracks, including a formidable *Long Text* challenge using GCoT. Through carefully designed prompts, it utilizes advanced vision-language models for nuanced human-aligned assessment of prompt-image alignment and image aesthetics. Our extensive evaluation of 19 leading models on PRISM-Bench reveals critical performance gaps and highlights specific areas requiring improvement. Our dataset, benchmark, and evaluation code are released at https://github.com/rongyaofang/prism-bench.

## 1 INTRODUCTION

Text-to-image generation models enable machines to produce engaging and coherent images, and have quickly become a key research direction in generative artificial intelligence (Ho et al., 2020; Liu et al., 2022; Rombach et al., 2022; Wu et al., 2022; Liang et al., 2022; OpenAI, September 2023; Podell et al., 2023; Chen et al., 2023b; Fang et al., 2024; Duan et al., 2025; Esser et al., 2024; Li et al., 2024c; BlackForest, 2024; Gong et al., 2025; Gao et al., 2025; Cai et al., 2025; Google, 2025b; OpenAI, 2025b; Wu et al., 2025; Google, 2025c). Among these models, state-of-the-art closed-source models (e.g., Gemini2.5-Flash-Image (Google, 2025c), GPT-Image-1 (OpenAI, 2025b)) demonstrate strong instruction following and controllable synthesis capabilities, establishing new benchmarks for T2I generation. In contrast, open-source models (Podell et al., 2023; Chen et al., 2023b; Stability-AI, 2022; 2024a;b; Chen et al., 2025a; Xie et al., 2025) exhibit limitations when processing complex and detailed prompts.

This disparity stems from two challenges. First, the research community lacks large-scale, high-quality, and comprehensive open-source datasets. Most existing datasets consist of web-crawled image-text pairs (Changpinyo et al., 2021; Schuhmann et al., 2022; Sharma et al., 2018; Hu et al., 2022; Gadre et al., 2023). These data are unable to be used to endow T2I models *reasoning* capabilities, which is the key for synthesizing complex scenes. Although reasoning-oriented datasets exist, they tend to be narrow in scope (Fang et al., 2025). For example, the GoT dataset (Fang et al., 2025) primarily focuses on layout planning through bounding boxes, offering limited coverage of other broader dimensions of reasoning. Second, there is an absence of a comprehensive evaluation

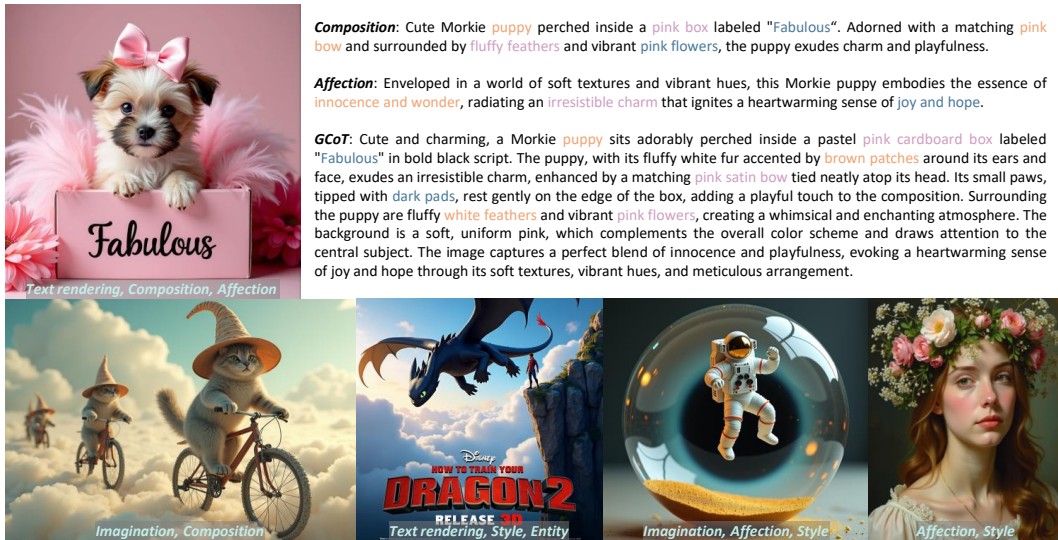

Figure 1: Showcase of FLUX-Reason-6M in six different characteristics and generation chain of thought. Keywords related to characteristics in the captions are highlighted in color.

benchmark aligned with human judgment. Most existing benchmarks (Ghosh et al., 2023; Huang et al., 2023; Hu et al., 2023; Chefer et al., 2023; Bakr et al., 2023; Feng et al., 2022; Yu et al., 2022; Wu et al., 2024; Cho et al., 2023) evaluate only a limited number of dimensions while neglecting key aspects such as imaginative capacity and emotional expression. Additionally, these benchmarks rely on object detectors (Ghosh et al., 2023) and crude CLIP scores (Hessel et al., 2021; Lin et al., 2024), resulting in evaluation metrics that are easily saturated and fail to effectively differentiate the model's actual performance.

To resolve these problems, in this work, we introduce **FLUX-Reason-6M** and **PRISM-Bench**. FLUX-Reason-6M is a 6-million-scale synthesized dataset designed to incorporate reasoning capabilities into the architecture of T2I generation. PRISM-Bench serves as a comprehensive and discriminative benchmark with 7 independent tracks that closely align with human judgment.

To build FLUX-Reason-6M, we leverage the powerful capabilities of advanced image generation models and vision-language models to develop a robust data pipeline that includes large-scale data collection, synthesis, mining, annotation, filtering, and translation. We identify six key characteristics essential for T2I generation: *Text rendering* (typography and legibility) and *Composition* (layout and spatial relations), which are common in existing researches (Huang et al., 2023; Wang et al., 2025; Chen et al., 2025b; 2023a; Tuo et al., 2023), and introduce *Imagination* (creative conceptual blending), *Affection* (emotional expression), *Entity* (knowledge-grounded depiction), and *Style* (artistic and photographic style) to capture more nuanced and creative aspects of generation. Furthermore, we introduce generation chain of thought (**GCoT**), which forms the core of our dataset. GCoT are detailed descriptions that break down the content and structure of images by comprehensively integrating the six characteristics instead of merely focusing on layout planning, providing supervision for training the reasoning capabilities of T2I models. As a result, FLUX-Reason-6M contains 6 million high-quality images synthesized with FLUX.1-dev (BlackForest, 2024) and 20 million associated captions, each in both English and Chinese. On average, each image contains at least three annotations from different categories. The creation of this dataset takes 15,000 A100 GPU days. Examples from the dataset can be found in Figure 1.

Building on the six characteristics and GCoT, we design PRISM-Bench. We first organize it into seven distinct tracks: the six categories from FLUX-Reason-6M and a uniquely challenging *Long Text* track that leverages the GCoT captions to test models' complex instruction following ability. Each track contains 100 carefully selected and constructed prompts. We leverage the sophisticated cognitive judgment of advanced vision-language models (GPT-4.1 (OpenAI, 2025a) and Qwen2.5-VL-72B (Bai et al., 2025)) to evaluate prompt-image alignment and aesthetic quality, thereby pro-

viding a more reliable and human-aligned assessment of model performance. We evaluate 19 leading T2I models, including SOTA closed-source models such as Gemini2.5-Flash-Image (Google, 2025c), GPT-Image-1 (OpenAI, 2025b), as well as top open-source models like Qwen-Image (Wu et al., 2025). The results indicate that the gap between open-source and closed-source models is widening, but even the most advanced closed-source models still have room for improvement in certain dimensions.

Our contributions are summarized as follows:

- We present **FLUX-Reason-6M**, a 6M-scale dataset for reasoning-oriented T2I, featuring 20M bilingual captions and, to our knowledge, the first multi-dimensional, million-scale *generation chain-of-thought* annotations tailored to controllable image synthesis.

- We introduce **PRISM-Bench**, a seven-track benchmark that evaluates *Imagination*, *Entity*, *Text rendering*, *Style*, *Affection*, and *Composition*, using GPT-4.1 and Qwen2.5-VL-72B as judges for nuanced and robust assessment.

- We conduct a comprehensive evaluation of 19 leading models, revealing revealing key gaps across characteristics and outlining concrete opportunities for future research.

- We will publicly release the dataset, benchmark, and evaluation suite to lower the financial and computational barriers, enabling research on reasoning-capable generative models.

## 2 FLUX-REASON-6M DATASET

The core limitation of current open-source T2I datasets is the absence of structured signals for complex reasoning (Podell et al., 2023; Li et al., 2024b; Stability-AI, 2024a). Most are flat image–caption collections that describe content but not the compositional rationale. We design FLUX-Reason-6M as a principled framework for learning T2I reasoning: beyond high-quality images, it supplies structured supervision that teaches the *rules* of composition and control. An overview of the curation pipeline appears in Figure 2.

### 2.1 SIX CHARACTERISTICS AND GENERATION CHAIN-OF-THOUGHT

We identify and define six key characteristics that are crucial for modern T2I models, including *Imagination*, *Entity*, *Text rendering*, *Style*, *Affection*, and *Composition*. These categories intentionally overlap to reflect the multifaceted nature of complex scene synthesis. For example, "The Eiffel Tower rendered in the style of Van Gogh's Starry Night" requires both *Entity* fidelity and *Style* transfer. At the core of the dataset is the *generated chain of thought*. As shown in Figure 1, GCoT breaks down semantic intent and compositional logic into multiple steps, providing rich supervision that teaches models not only vocabulary-pixel associations but also underlying rules for layout, typography, emotional tones, and stylistic choices. This GCoT-centered multi-dimensional framework forms the conceptual foundation of FLUX-Reason-6M. Detailed definitions are in appendix B.1.

### 2.2 SYNTHESIZING A HIGH-QUALITY VISUAL FOUNDATION

To avoid the uneven quality of web data, we select the powerful FLUX.1-dev (BlackForest, 2024) as our synthesis engine, which produces detailed and aesthetically consistent images. We first rewrite captions from LAION-Aesthetics (Schuhmann et al., 2022) using a vision language model to obtain high quality, broad descriptions as starting points. However, this strategy underrepresents *Imagination* and *Text rendering*, thus we implement targeted augmentation.

**Progressive Imagination Cultivation.** We initiate a progressive generation process. First, we use Gemini-2.5-Pro (Google, 2025a) to generate 200 imaginative seed prompts. Then we randomly select 10 of these prompts, input them as in-context examples to Qwen3-32B (Yang et al., 2025) for expansion, and increase the model's temperature parameter to enhance novelty and diversity.

**Mining-Generation-Synthesis Pipeline for Textual Rendering.** We develop a three-stage pipeline. First, we mine Laion-2B (Schuhmann et al., 2022) with Qwen2.5-VL-32B (Bai et al., 2025) to find images with clear, legible text. For each hit we generate precise captions that describe

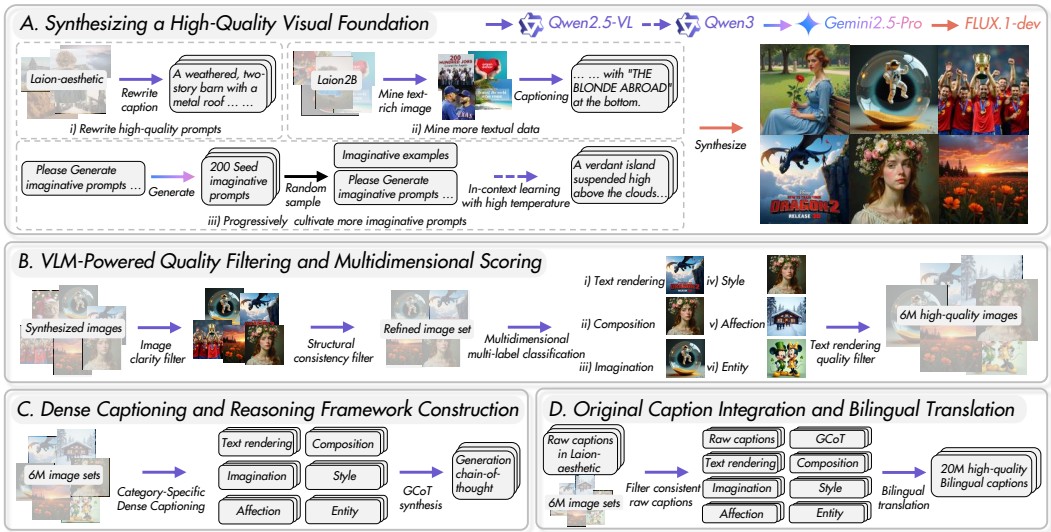

Figure 2: An overview of FLUX-Reason-6M data curation pipeline. The entire process was completed using 128 A100 GPUs over a period of 4 months.

the text content, visual presentation, and context. We then synthesize images with FLUX.1-dev so that the rendered text matches the caption.

Combining the baseline and augmentations yields 8M images for subsequent filtering, multidimensional categorization, and dense annotation, ensuring a consistent standard of quality and relevance for FLUX-Reason-6M.

## 2.3 VLM-POWERED QUALITY FILTERING AND MULTIDIMENSIONAL SCORING

We convert 8 million synthesized images into a carefully curated resource using a pipeline powered by vision language models. The goal is to ensure high visual quality and reliable alignment with the six characteristics. We first apply a basic quality screen with Qwen-VL to remove blur, artifacts, and structural errors. The full rubric and thresholds are in appendix B.2. Then we again use Qwen-VL to score each remaining image on all six characteristics separately. The model assigns relevance scores from 1 to 10 for each category. We calibrate thresholds for each characteristic and assign categories accordingly. When an image meets multiple thresholds, it is assigned multiple labels. For the *Text rendering* category, we add a specialized pass that rejects images with illegible, low contrast, or incorrect text. Implementation details are provided in appendix B.3. From the original 8 million candidates, about 6 million pass all checks. These images are quality validated and carry multidimensional labels, and they are used for the final dense annotation stage.

## 2.4 VLM-DRIVEN DENSE CAPTIONING AND REASONING FRAMEWORK CONSTRUCTION

With the filtered and classified images in place, we generate multidimensional captions and construct a reasoning framework. The goal is to move beyond simple descriptions and create annotations that explain what is in the scene and how it is organized.

**Category-Specific Dense Captioning.** Using Qwen-VL, we create category aware captions for each assigned characteristic. For *Entity*, the caption prioritizes precise identification and attributes. For *Style*, it describes techniques, aesthetics, and visual motifs. For *Text rendering*, it captures typography and layout. The same principle applies to *Imagination*, *Affection*, and *Composition*. Images with multiple labels receive parallel captions from each perspective, which yields denser supervision than generic captioning methods (Li et al., 2020; Jia et al., 2021; Yu et al., 2024).

**Generation Chain-of-Thought Synthesis.** We then build GCoT, a central feature of FLUX-Reason-6M. The VLM receives the image and all category specific captions and returns a detailed

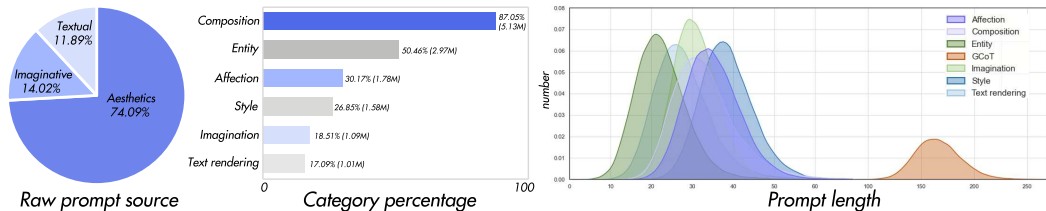

Figure 3: **Left**: Three subsets of raw prompt sources. **Middle**: Image category ratio. **Right**: Prompt Suite Statistics.

plan that explains scene elements, their interactions, layout choices, and guiding principles. These narratives cover spatial relations, color and style decisions, typography quality, and emotional tone. The result is an explicit template for reasoning and provides structured signals for training.

## 2.5 ORIGINAL CAPTION INTEGRATION AND BILINGUAL RELEASE AT SCALE

**Original Caption Integration.** We reintegrate high quality captions from LAION-Aesthetics (Schuhmann et al., 2022) when they match the FLUX generated images. Qwen-VL serves as an alignment judge that scores caption-image alignment. Captions above a calibrated threshold are kept to expand linguistic variety while avoiding drift. After merging original captions, category specific captions, and GCoT annotations, the corpus contains about 20 million unique captions. Figure 3 summarizes source proportions, counts and shares of caption types, and word count distributions.

**Comprehensive Bilingual Translation.** We translate the entire caption set into Chinese using VLM. For the *Text rendering* category, any English string that must appear in the image remains in English to preserve the task meaning. The result is a bilingual resource that supports broad use across regions and applications.

## 3 PRISM-BENCH

To address the evaluation gap in T2I synthesis, we introduce PRISM-Bench. It consists of seven distinct tracks, each containing 100 carefully selected prompts designed to explore the capability boundaries of T2I models. These tracks align with our six characteristics, namely *Imagination*, *Entity*, *Style*, *Text rendering*, *Composition*, and *Affection*, and include a challenging *Long Text* built from GCoT prompts. An overview of our PRISM-Bench is shown in Figure 4.

### 3.1 PROMPT DESIGN AND CONSTRUCTION

Each track contains 100 prompts split into two sets of 50. The first set is representative samples from FLUX-Reason-6M. The second set targets hard cases within the track.

**Representative Prompt Sampling.** For each track, we collect the top 10,000 prompts based on the scores in section 2.3. We use k-means (Krishna & Murty, 1999) for semantic clustering (k=50) and select prompts closest to each centroid, removing them from the dataset. In this way, we obtain 50 prompts that cover major themes and reduce bias toward frequent patterns.

**Category-Specific Prompt Construction.** The other 50 prompts for each track come from our careful curation. For *Text rendering*, we define pools for content length, typography, and placement context. Content ranges from a single word to multi-sentence slogans. Typography spans handwriting, serif, sans serif, and graffiti. Placement covers signs, garments, packaging, and other surfaces, with explicit constraints on viewpoint, lighting, and contrast. We sample combinations from these pools and ask Gemini2.5-Pro to compose natural prompts, with checks for legibility and string accuracy. The other tracks, including *Imagination*, *Entity*, *Style*, *Affection*, *Composition*, and *Long Text*, follow the similar principle and are detailed in appendix C.1.

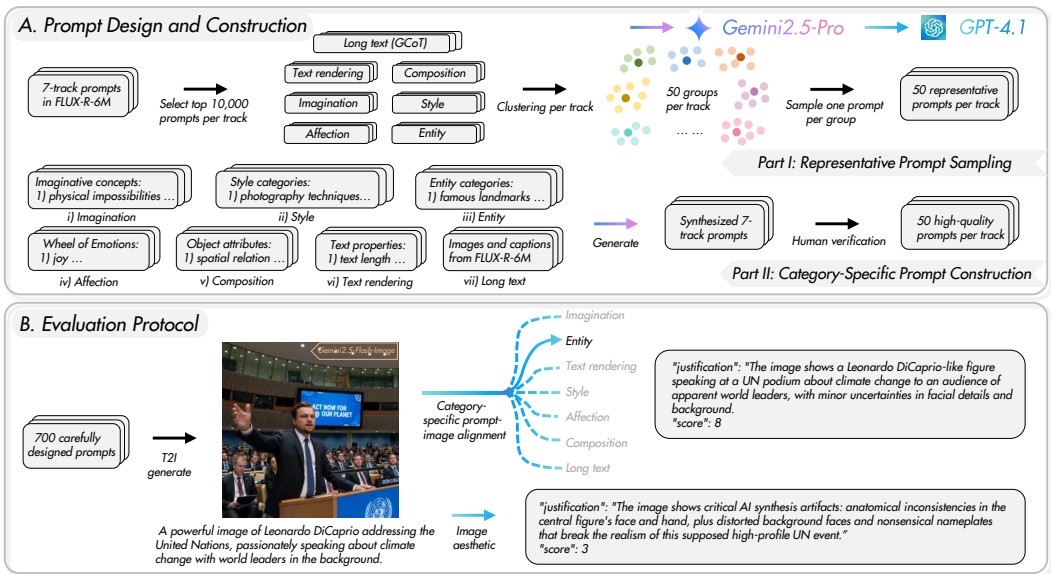

Figure 4: An overview of the prompt design and evaluation protocol of PRISM-Bench.

**PRISM-Bench-ZH.** We construct PRISM-Bench-ZH by translating all English prompts into Chinese with Gemini2.5-Pro. For *Text rendering* track, we do not simply translate all text into Chinese but adapt it according to Chinese contexts. For example, "A bottle labeled 'WHISTLEPIG' featuring 'SMOKED BARREL-AGED RYE' sits alongside two clear whiskey glasses, showcasing a refined presentation of the spirit" is translated as "一个标有'茅台'并写着'珍品酱香型白酒'的酒瓶，旁边放着两个透明的白酒杯，尽显这款烈酒的精致典雅。"

**Human-in-the-Loop Refinement.** Five human reviewers participate in the validation of prompts: two senior PhD candidates (with 3-4 years of experience in AIGC), one master's student (with expertise in human-computer interaction research), one graphic designer (possessing over 5 years of experience in digital art and visual communication), and one postdoctoral researcher (with extensive research background in AIGC). All the reviewers are instructed to evaluate each candidate prompt on grammatical correctness and clarity, logical consistency of the described scene, feasibility for T2I generation, absence of ambiguity in spatial/attribute descriptions, alignment with the specific track's focus, absence of duplication with existing prompts, and natural language flow and creative coherence. Only prompts approved by all reviewers entered the final benchmark. Prompts with disagreements were revised iteratively or discarded. The final benchmark contains 700 prompts that are diverse, representative, challenging, and bilingual.

## 3.2 EVALUATION PROTOCOL

We use vision language models as judges to assess prompt–image alignment and image aesthetics. This yields a two-axis evaluation that captures both correctness and visual quality. We employ GPT-4.1 and Qwen2.5-VL-72B as representative closed source and open source judges.

**Fine-Grained Alignment Evaluation.** Alignment scoring uses track-specific instructions that direct judges to focus on the core challenges of each track. For each generated image, the judge provides a one-sentence explanation and assigns a score from 1 to 10. Detailed criteria for all tracks are provided in appendix C.2.

**Uniform Aesthetic Evaluation.** Aesthetic scoring adopts a unified evaluation standard across all tracks. The VLM considers factors such as lighting, color harmony, detail rendering, and overall visual appeal, providing a one-sentence rationale along with a numerical score ranging from 1 to 10.

For each model and track, we average the alignment and aesthetic scores across 100 prompts, mapping them to a scale of 0 to 100, and report their mean as the comprehensive track score. The

Table 1: Quantitative results on PRISM-Bench evaluated by GPT-4.1. Ali., Aes., and Avg. denote alignment, aesthetic, and average scores, respectively. The best result is in bold and the second best result is underlined.

| Model | Imagination | | | Entity | | | Text rendering | | | Style | | | Affection | | | Composition | | | Long text | | | Overall | | |
|---|---|---|---|---|---|---|---|---|---|---|---|---|---|---|---|---|---|---|---|---|---|---|---|---|
| | Ali. | Aes. | Avg. | Ali. | Aes. | Avg. | Ali. | Aes. | Avg. | Ali. | Aes. | Avg. | Ali. | Aes. | Avg. | Ali. | Aes. | Avg. | Ali. | Aes. | Avg. | Ali. | Aes. | Avg. |
| SD1.5 | 36.6 | 36.1 | 36.4 | 53.8 | 41.1 | 47.5 | 8.0 | 33.1 | 20.6 | 55.3 | 55.3 | 55.3 | 64.4 | 57.5 | 61.0 | 61.1 | 51.0 | 56.1 | 35.3 | 30.4 | 32.9 | 44.9 | 43.5 | 44.2 |
| SD2.1 | 47.9 | 41.2 | 44.6 | 60.9 | 46.7 | 53.8 | 11.2 | 30.6 | 20.9 | 62.7 | 58.6 | 60.7 | 66.7 | 58.5 | 62.6 | 65.7 | 53.1 | 59.4 | 40.1 | 28.2 | 34.2 | 50.7 | 45.3 | 48.0 |
| SDXL | 55.3 | 61.1 | 58.2 | 72.5 | 67.4 | 70.0 | 13.8 | 37.0 | 25.4 | 72.4 | 75.4 | 73.9 | 78.9 | 77.1 | 78.0 | 75.5 | 75.3 | 75.4 | 44.2 | 39.6 | 41.9 | 58.9 | 61.8 | 60.4 |
| JanusPro-7B | 70.4 | 65.8 | 68.1 | 67.1 | 51.9 | 59.5 | 15.5 | 36.7 | 26.1 | 71.4 | 73.8 | 72.6 | 79.2 | 71.5 | 75.4 | 83.7 | 61.0 | 72.4 | 62.4 | 39.7 | 51.1 | 64.2 | 57.2 | 60.7 |
| Playground | 62.3 | 70.6 | 66.5 | 72.5 | 69.1 | 70.8 | 10.4 | 37.3 | 23.9 | 77.3 | 80.9 | 79.1 | 91.8 | 83.8 | 87.8 | 77.5 | 76.5 | 77.0 | 46.7 | 41.0 | 43.9 | 62.6 | 65.6 | 64.1 |
| FLUX.1-schnell | 63.3 | 66.2 | 64.8 | 61.8 | 51.2 | 56.5 | 46.2 | 54.1 | 50.2 | 68.6 | 70.1 | 69.4 | 75.4 | 69.9 | 72.7 | 85.1 | 67.5 | 76.3 | 69.4 | 49.7 | 59.6 | 67.1 | 61.2 | 64.2 |
| Bagel | 69.4 | 68.0 | 68.7 | 59.0 | 50.1 | 54.6 | 30.2 | 44.5 | 37.4 | 67.9 | 71.3 | 69.6 | 81.7 | 81.4 | 81.6 | 90.5 | 73.1 | 81.8 | 68.1 | 55.3 | 61.7 | 66.7 | 63.4 | 65.1 |
| Bagel-CoT | 68.4 | 74.2 | 71.3 | 62.4 | 60.0 | 61.2 | 23.2 | 40.1 | 31.7 | 64.4 | 70.1 | 67.3 | 87.1 | 80.5 | 83.8 | 88.5 | 77.9 | 83.2 | 64.0 | 52.0 | 58.0 | 65.4 | 65.0 | 65.2 |
| SD3-Medium | 61.0 | 65.6 | 63.3 | 64.8 | 56.3 | 60.6 | 32.8 | 53.1 | 43.0 | 74.8 | 75.6 | 75.2 | 78.7 | 80.3 | 79.5 | 85.5 | 79.1 | 82.3 | 61.5 | 46.1 | 53.8 | 65.6 | 65.2 | 65.4 |
| SD3.5-Medium | 69.5 | 73.0 | 71.3 | 72.8 | 63.7 | 68.3 | 33.3 | 50.1 | 41.7 | 77.4 | 80.3 | 78.9 | 84.9 | 85.5 | 85.2 | 89.4 | 79.2 | 84.3 | 63.3 | 50.5 | 56.9 | 70.1 | 68.9 | 69.5 |
| HiDream-I1-Dev | 68.2 | 69.7 | 69.0 | 72.0 | 67.0 | 69.5 | 53.4 | 64.1 | 58.8 | 68.7 | 78.6 | 73.7 | 84.2 | 83.1 | 83.7 | 87.6 | 79.8 | 83.7 | 58.1 | 47.5 | 52.8 | 70.3 | 70.0 | 70.2 |
| SD3.5-Large | 73.3 | 71.2 | 72.3 | 76.7 | 71.9 | 74.3 | 52.0 | 65.8 | 58.9 | 77.1 | 84.2 | 80.7 | 87.1 | 85.2 | 86.2 | 87.0 | 84.7 | 85.9 | 64.3 | 51.7 | 58.0 | 73.9 | 73.5 | 73.7 |
| FLUX.1-dev | 68.1 | 74.0 | 71.1 | 70.7 | 71.2 | 71.0 | 48.1 | 64.5 | 56.3 | 72.3 | 80.5 | 76.4 | 88.3 | 91.1 | 89.7 | 89.0 | 84.6 | 86.8 | 70.6 | 58.5 | 64.6 | 72.4 | 74.9 | 73.7 |
| FLUX.1-Krea-dev | 71.5 | 73.0 | 72.3 | 69.5 | 67.5 | 68.5 | 47.5 | 61.3 | 54.4 | 80.8 | 83.5 | 82.2 | 84.0 | 90.3 | 87.2 | 90.9 | 85.8 | 88.4 | 76.2 | 64.1 | 70.2 | 74.3 | 75.1 | 74.7 |
| HiDream-I1-Full | 74.4 | 75.6 | 75.0 | 74.4 | 72.4 | 73.4 | 58.2 | 70.4 | 64.3 | 81.4 | 84.8 | 83.1 | 90.1 | 88.8 | 89.5 | 90.1 | 85.4 | 87.8 | 63.8 | 52.0 | 57.9 | 76.1 | 75.6 | 75.9 |
| SEEDream 3.0 | 77.3 | 76.4 | 76.9 | 80.2 | 73.8 | 77.0 | 56.1 | 70.2 | 63.2 | 83.9 | 87.4 | 85.7 | 89.3 | 90.3 | 89.8 | 93.3 | 86.3 | 89.8 | 83.2 | 66.7 | 75.0 | 80.5 | 78.7 | 79.6 |
| Qwen-Image | 80.5 | 78.6 | 79.6 | 79.3 | 73.2 | 76.3 | 54.3 | 68.9 | 61.6 | 84.5 | 88.7 | 86.6 | 91.6 | 89.1 | 90.4 | 93.7 | 86.9 | 90.3 | 83.8 | 65.1 | 74.5 | 81.1 | 78.6 | 79.9 |
| Gemini2.5-Flash-Image | 92.4 | 84.8 | 88.6 | 87.0 | 81.3 | 84.2 | 65.2 | 74.1 | 69.7 | 90.5 | 90.8 | 90.7 | 96.0 | 88.2 | 92.1 | 92.5 | 88.5 | 90.5 | 85.9 | 76.2 | 81.1 | 87.1 | 83.4 | 85.3 |
| GPT-Image-1 [High] | 86.2 | 86.6 | 86.4 | 90.0 | 86.3 | 88.2 | 68.8 | 80.1 | 74.5 | 92.8 | 93.3 | 93.1 | 90.7 | 90.9 | 90.8 | 96.2 | 89.4 | 92.8 | 83.8 | 72.8 | 78.3 | 86.9 | 85.6 | 86.3 |

final model score is calculated as the average across all seven tracks. We evaluate leading closed-source and open-source T2I systems using both PRISM-Bench and PRISM-Bench-ZH to facilitate a comprehensive comparison.

# 4 EXPERIMENTS

We evaluate 19 advanced image generation models on the PRISM-Bench, including Gemini2.5-Flash-Image (Google, 2025c), GPT-Image-1 (OpenAI, 2025b), Qwen-Image (Wu et al., 2025), SEEDream 3.0 (Gao et al., 2025), FLUX series (BlackForest, 2024; 2025), HiDream series (Cai et al., 2025), Stable Diffusion series (Rombach et al., 2022; Podell et al., 2023; Stability-AI, 2022; 2024a;b), Playground (Li et al., 2024b), Bagel (Deng et al., 2025), and JanusPro (Chen et al., 2025c). Results under the GPT-4.1 judge appear in Table 1. The corresponding Qwen-VL judge results are provided in appendix A. Meanwhile, we evaluate several models with Chinese language capabilities on the PRISM-Bench-ZH, including GPT-Image-1, Qwen-Image, SEEDream 3.0, HiDream series, and Bagel. Table 2 reports results, and the Qwen-VL results are in appendix A.

## 4.1 RESULTS AND ANALYSIS ON PRISM-BENCH

As shown in Table 1, the overall results highlight the advantages of current SOTA closed-source models. GPT-Image-1 achieves the highest total score of 86.3, closely followed by Gemini2.5-Flash-Image with 85.3. These models outperform others across nearly all evaluation tracks. Among the remaining models, a competitive tier led by Qwen-Image is emerging. Although there is still a noticeable performance gap compared to top models, these models represent a significant leap forward from the open-source community. HiDream-I1-Full and FLUX.1-Krea-dev also achieve excellent results, indicating rapid progress in the field. Evolution within model series is also evident, with SDXL showing substantial improvement over SD1.5, while the newer SD3.5-Large further narrows the gap with top-performing models. Among all tracks, *Style* and *Composition* are relatively mature domains, whereas *Text rendering* and *Long text* remain the most challenging. *Imagination* and *Entity* recognition also distinguish top models by their emphasis on creative synthesis and factual accuracy. Figure 5 presents a representative *Long text* case, illustrating the difficulty in adhering

*From a high vantage point, a person's left hand holds a black smartphone in the foreground, creating a clever frame-within-a-frame effect. The screen of the phone displays a crystal-clear image of the road below, perfectly aligned with the real-world scene, making it seem as though one is looking through a transparent window rather than at a device. The road, flanked by a dense, lush canopy of dark green trees that form a natural tunnel, stretches into the distance. Several cars, including black, white, and a distinct red vehicle, are visible driving along the multi-lane asphalt. A curved, modern street lamp extends over the roadway from the left, nestled amongst the foliage. The photograph is shot with a shallow depth of field, causing the hand in the immediate foreground and the distant trees to be softly blurred, drawing the viewer's eye to the sharp focus on the phone and the road just beyond it. The overall color palette is moody and cinematic, with cool, desaturated tones emphasizing the deep greens of the trees and the dark grey of the road, which is marked with white painted lines.*

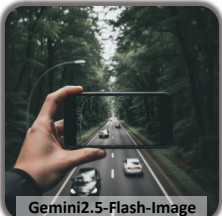

**Gemini2.5-Flash-Image**

**Justification:** *Accurately shows major elements (high vantage, left hand, black smartphone, frame-within-a-frame, road with cars, dense green canopy, modern street lamp, shallow depth). Minor deduction for imperfect phone screen alignment and only moderate cinematic desaturation.*
***Alignment score*: 9**

**Justification:** *Frame-within-a-frame is convincing, but slight unnatural blending at street lamp/tree canopy transition and hand appears stiff with less skin texture. Minor flaws, overall realism maintained.*
***Aesthetic score*: 8**

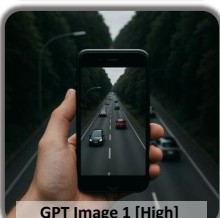

**GPT Image 1 [High]**

**Justification:** *All major elements present (high vantage, left hand with black smartphone, frame-within-a-frame, road with cars including red one, dense green canopy, shallow depth, moody palette). Minor deduction: street lamp only partially visible, not clearly among foliage.*
***Alignment score: 9***

**Justification:** *Strong frame-within-a-frame and realism, but major flaw: phone screen and background misaligned (road perspective, street lamp position), breaking the transparent window illusion; hand anatomy off (unnatural thumb curve). Deductions for logical and anatomical issues.*
***Aesthetic score*: 5**

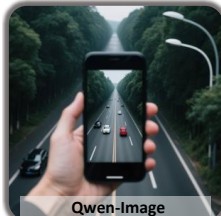

**Qwen-Image**

**Justification:** *Critical flaw: smartphone screen misaligned with real scene—road and vehicles don't match, breaking transparency illusion. Hand slightly distorted; street lamp base awkwardly blended with foliage, reducing realism.*
***Alignment score: 9***

**Justification:** *All key elements present (high vantage, left hand, black smartphone, frame-within-frame, road, specified car colors, tree tunnel, street lamp, shallow depth, moody palette). Minor issues: street lamp not clearly modern/curved; desaturation only moderate.*
***Aesthetic score*: 2**

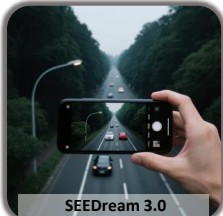

**SEEDream 3.0**

**Justification:** *Strong frame-within-a-frame alignment and depth of field, but minor hand-phone blending issues: fingers are slightly stiff and thumb position is subtly awkward, affecting anatomical realism.*
***Alignment score: 6***

**Justification:** *Captures all major elements (high vantage, black smartphone, frame-within-a-frame, road, cars, dense canopy, modern lamp, shallow DOF, moody palette), but the hand is not clearly identifiable as a left hand and cinematic, cool, desaturated grading is only moderately present.*
***Aesthetic score*: 6**

Figure 5: Showcase of *Long text* track in the PRISM-Bench. GPT4.1 is not only required to score based on image-text alignment and image aesthetics, but also to provide a brief justification.

to dense instructions. The Qwen-VL evaluation results yields consistent rankings, as detailed in appendix A.

## 4.2 RESULTS AND ANALYSIS ON PRISM-BENCH-ZH

The evaluation results from PRISM-Bench-ZH reveal a distinct performance hierarchy, with GPT-Image-1 establishing its dominance at a total score of 87.5. It consistently leads across most tracks, including *Imagination*, *Entity*, *Style*, *Affection*, and *Composition*, demonstrating exceptional creative interpretation, knowledge foundation, and spatial arrangement in response to Chinese prompts. Meanwhile, SEEDream 3.0 and Qwen-Image demonstrate strong competitiveness across all tracks, frequently performing nearly on par with the leader. Particularly noteworthy is the performance of SEEDream 3.0 and Qwen-Image in *Text rendering*, which stands in stark contrast to the general weakness observed in English text generation. Among these, SEEDream 3.0 and GPT-Image-1 share the highest average score, with SEEDream 3.0 achieving the highest aesthetic score, indicating its capability to render high-quality Chinese characters. The robust performance of these models validates the benchmark design's use of culturally adaptive prompts in Chinese and highlights signif-

Table 2: Quantitative results on PRISM-Bench-ZH evaluated by GPT-4.1. The best result is in bold and the second best result is underlined.

| Model | Imagination | | | Entity | | | Text rendering | | | Style | | | Affection | | | Composition | | | Long text | | | Overall | | |
|---|---|---|---|---|---|---|---|---|---|---|---|---|---|---|---|---|---|---|---|---|---|---|---|---|
| | Ali. | Aes. | Avg. | Ali. | Aes. | Avg. | Ali. | Aes. | Avg. | Ali. | Aes. | Avg. | Ali. | Aes. | Avg. | Ali. | Aes. | Avg. | Ali. | Aes. | Avg. | Ali. | Aes. | Avg. |
| HiDream-I1-Dev | 47.3 | 41.1 | 44.2 | 52.8 | 49.0 | 50.9 | 35.2 | 14.5 | 24.9 | 64.5 | 52.4 | 58.5 | 76.3 | 66.5 | 71.4 | 67.6 | 68.3 | 68.0 | 41.1 | 46.4 | 43.8 | 55.0 | 48.3 | 51.7 |
| HiDream-I1-Full | 53.6 | 47.3 | 50.5 | 63.1 | 60.8 | 62.0 | 34.6 | 16.3 | 25.5 | 74.1 | 65.5 | 69.8 | 80.9 | 67.3 | 74.1 | 73.8 | 76.1 | 75.0 | 45.4 | 50.8 | 48.1 | 60.8 | 54.9 | 57.9 |
| Bagel-CoT | 75.1 | 69.3 | 72.2 | 53.3 | 58.8 | 56.1 | 42.6 | 16.3 | 29.5 | 73.6 | 66.6 | 70.1 | 81.2 | 78.0 | 79.6 | 74.0 | 83.6 | 78.8 | 50.7 | 64.3 | 57.5 | 64.4 | 62.4 | 63.4 |
| Bagel | 72.8 | 64.7 | 68.8 | 53.9 | 62.2 | 58.1 | 49.2 | 29.0 | 39.1 | 73.9 | 68.4 | 71.2 | 81.4 | 73.5 | 77.5 | 69.0 | 89.8 | 79.4 | 58.1 | 68.7 | 63.4 | 65.5 | 65.2 | 65.4 |
| Qwen-Image | 80.1 | 79.6 | 79.9 | 75.6 | 79.7 | 77.7 | 76.9 | 62.9 | 69.9 | 90.2 | 84.3 | 87.3 | 87.4 | 84.9 | 86.2 | 86.6 | 93.4 | 90.0 | 68.9 | 84.2 | 76.6 | 80.8 | 81.3 | 81.1 |
| SEEDream 3.0 | 77.2 | 77.8 | 77.5 | 77.6 | 78.6 | 78.1 | 79.7 | 71.9 | 75.8 | 87.8 | 83.2 | 85.5 | 88.7 | 85.1 | 86.9 | 87.7 | 94.4 | 91.1 | 74.3 | 82.7 | 78.5 | 81.9 | 82.0 | 82.0 |
| GPT-Image-1 [High] | 88.8 | 90.4 | 89.6 | 85.9 | 92.4 | 89.2 | 83.9 | 67.7 | 75.8 | 93.9 | 91.7 | 92.8 | 91.5 | 86.5 | 89.0 | 92.4 | 97.3 | 94.9 | 77.2 | 84.3 | 80.8 | 87.7 | 87.2 | 87.5 |

icant advancements in handling Chinese typography. Nevertheless, consistent with PRISM-Bench test results, the *Long text* track remains the greatest challenge for all models. While GPT-Image-1 again leads in this category, the generally lower scores highlight the substantial obstacle of understanding and synthesizing lengthy, multifaceted Chinese instructions. This further emphasizes the urgent need for reasoning-focused datasets like FLUX-Reason-6M to address existing gaps and train the next generation of truly intelligent T2I models. The Qwen-VL judge results and visualizations for PRISM-Bench-ZH are included in appendix A.

## 4.3 HUMAN EVALUATION ON PRISM-BENCH

We further conducted a comprehensive human evaluation on PRISM-Bench and compared human scores with our automatic metrics. We randomly sampled 20 prompts from each of the 7 tracks and generated images using 4 diverse T2I models (BAGEL, FLUX.1-dev, Qwen-Image, Gemini2.5-Flash-Image), yielding 560 image-prompt pairs. We recruited 10 well-educated graduate students to conduct human evaluation. These evaluators represented diverse disciplinary backgrounds including arts, design, and STEM fields. Each image was randomly assigned to three evaluators who conducted independent evaluations. Evaluators were instructed to rate each image according to the same prompts provided to the VLM, assessing both prompt–image alignment and aesthetic quality. They then provided a composite score (1-10) for each image based on these criteria. We averaged the three scores to obtain a human reference score for each prompt-image pair. We computed Spearman's $\rho$ and Kendall's $\tau$ between human scores and three automatic evaluators, including CLIPScore (Hessel et al., 2021), our VLM-based scores using Qwen2.5-VL-72B, and our VLM-based scores using GPT-4.1. We also calculated inter-judge correlation between GPT-4.1 and Qwen2.5-VL-72B across the average scores of all 19 models on each track. The results are shown in Table 3. Our VLM-based metrics achieve substantially higher correlation with human judgment than CLIPScore across all tracks, validating PRISM-Bench as a more human-aligned evaluation standard than traditional metrics. Besides, the consistently high correlation ($\rho > 0.86$ across all tracks) demonstrates remarkable agreement between the two judges despite their different architectures and training. This validates the robustness and reliability of PRISM-Bench's evaluation protocol.

## 5 RELATED WORKS

### 5.1 DATASETS FOR T2I

Large-scale web-scraped corpora (Changpinyo et al., 2021; Schuhmann et al., 2022; Sharma et al., 2018; Hu et al., 2022; Gadre et al., 2023) form the foundation of most open-source T2I models. These datasets provide broad coverage across domains, but at the cost of heterogeneous quality and weak alignment signals. While such datasets describe *what* is in an image, they typically lack structured supervision about *how* or *why* the content is composed, thus limiting the ability to train reasoning-oriented models. To partially address this issue, the GoT dataset (Fang et al., 2025) scales 9M samples and introduces reasoning elements for composition via bounding box layouts. However, this dataset is primarily assembled from existing sources (e.g., Laion-Aesthetics (Schuhmann et al., 2022), JourneyDB (Sun et al., 2023)), leading to inconsistent quality and imbalanced distributions

Table 3: The correlation between automatic evaluation metrics and human evaluation, and the correlation between the two metrics used in PRISM-Bnech (GPT-4.1 vs. Qwen2.5-VL).

| Metrics | Imagination | | Entity | | Text rendering | | Style | | Affection | | Composition | | Long text | |
|---|---|---|---|---|---|---|---|---|---|---|---|---|---|---|
| | $\rho$ | $\tau$ | $\rho$ | $\tau$ | $\rho$ | $\tau$ | $\rho$ | $\tau$ | $\rho$ | $\tau$ | $\rho$ | $\tau$ | $\rho$ | $\tau$ |
| CLIPScore | 0.415 | 0.306 | 0.371 | 0.292 | 0.645 | 0.498 | 0.427 | 0.318 | 0.356 | 0.283 | 0.698 | 0.502 | 0.223 | 0.154 |
| Ours (Qwen2.5-VL-72B) | 0.527 | 0.382 | 0.585 | 0.456 | 0.709 | 0.547 | 0.658 | 0.484 | 0.503 | 0.372 | 0.735 | 0.576 | 0.662 | 0.497 |
| Ours (GPT-4.1) | **0.580** | **0.453** | **0.626** | **0.481** | **0.722** | **0.569** | **0.694** | **0.531** | **0.559** | **0.397** | **0.741** | **0.585** | **0.683** | **0.510** |
| GPT4.1 vs. Qwen2.5-VL | 0.861 | 0.720 | 0.872 | 0.704 | 0.978 | 0.903 | 0.973 | 0.888 | 0.872 | 0.739 | 0.935 | 0.806 | 0.982 | 0.947 |

of image content and style, and its chain-of-thought remains largely layout-centric. In contrast, FLUX-Reason-6M is constructed through a unified synthesis and annotation pipeline, targeting six complementary features (*Imagination*, *Entity*, *Text rendering*, *Style*, *Affection*, *Composition*), and provides multi-aspect natural language generation chain of thought supervision. To our knowledge, large-scale datasets with such multi-dimensional, bilingual GCoT remain scarce in public literature.

## 5.2 EVALUATION AND BENCHMARKS OF T2I

Evaluating T2I models is challenging. CLIP-based metrics (e.g., CLIPScore (Hessel et al., 2021)) and detector-driven checks (Ghosh et al., 2023) have been widely used due to scalability, but can saturate and sometimes mis-rank outputs (Lin et al., 2024). Benchmarks like T2I-CompBench (Huang et al., 2023), TIFA (Hu et al., 2023), Conceptmix (Wu et al., 2024), and DSG (Cho et al., 2023) provide targeted coverage (e.g., attribute binding, spatial relations), yet often emphasize a limited set of dimensions and rely on automated proxies that do not fully reflect human preferences. A complementary direction is leveraging vision–language models as judges (Hu et al., 2024; Li et al., 2024a). Strong VLMs (e.g., GPT-4.1 (OpenAI, 2025a), Qwen2.5-VL (Bai et al., 2025)) can perform fine-grained, instruction-aware comparisons that better correlate with human assessments than pipelines using only detectors or CLIP, especially for nuanced aspects like creativity, emotional tone, and long-instruction compliance. PRISM-Bench follows this direction by organizing evaluation into six tracks aligned with our six characteristics plus a *Long text* track using GCoT prompts, and employing VLM judges to score prompt-image alignment and aesthetics.

## 6 CONCLUSION

In this work, we address critical gaps in text-to-image models through two key contributions: the FLUX-Reason-6M dataset and the PRISM benchmark. FLUX-Reason-6M is an extensive 6-million-image dataset with 20 million high-quality prompts specifically designed for reasoning, featuring novel generation chain-of-thought that imparts image synthesis logic across six characteristics to models. To measure progress, we develop PRISM-Bench, a comprehensive seven-track benchmark utilizing advanced VLMs for fine-grained human-aligned evaluation. Our extensive experimentation across 19 models reveals that while leading closed-source systems demonstrate impressive performance, all models struggle with complex tasks such as text rendering and long instruction following, underscoring the necessity of our work. By publicly releasing the dataset, benchmark, and evaluation code, we provide the community with essential tools for training and evaluating the next generation of more intelligent and capable T2I models.

## 7 REPRODUCIBILITY STATEMENT

We describe the dataset creation process in section 2, with additional details provided in appendix B. The benchmark construction is outlined in section 3, with further elaboration presented in appendix C. The prompts used for benchmark evaluation are exhibited in appendix C.2 and C.3. Our dataset, benchmark, and code will be publicly available.

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

Table 4: Quantitative results on PRISM-Bench evaluated by Qwen2.5-VL-72B. The best result is in bold and the second best result is underlined.

| Model | Imagination | | | Entity | | | Text rendering | | | Style | | | Affection | | | Composition | | | Long text | | | Overall | | |
|---|---|---|---|---|---|---|---|---|---|---|---|---|---|---|---|---|---|---|---|---|---|---|---|---|
| | Ali. | Aes. | Avg. | Ali. | Aes. | Avg. | Ali. | Aes. | Avg. | Ali. | Aes. | Avg. | Ali. | Aes. | Avg. | Ali. | Aes. | Avg. | Ali. | Aes. | Avg. | Ali. | Aes. | Avg. |
| SD1.5 | 40.7 | 23.7 | 32.2 | 61.2 | 52.7 | 56.9 | 11.4 | 24.1 | 17.8 | 56.7 | 61.5 | 59.1 | 66.9 | 60.7 | 63.8 | 57.5 | 53.4 | 55.4 | 47.3 | 26.8 | 37.0 | 48.8 | 43.3 | 46.0 |
| SD2.1 | 48.9 | 28.4 | 38.6 | 66.0 | 57.6 | 61.8 | 16.7 | 31.4 | 24.0 | 62.7 | 66.5 | 64.6 | 68.5 | 62.1 | 65.3 | 64.8 | 58.3 | 61.5 | 50.7 | 29.8 | 40.2 | 54.0 | 47.7 | 50.8 |
| SDXL | 54.5 | 34.1 | 44.3 | 71.1 | 65.0 | 68.0 | 18.6 | 37.3 | 27.9 | 71.7 | 72.6 | 72.1 | 78.7 | 66.5 | 72.6 | 72.2 | 67.8 | 70.0 | 54.1 | 34.5 | 44.3 | 60.1 | 54.0 | 57.0 |
| Playground | 59.0 | 39.0 | 49.0 | 69.4 | 56.7 | 63.0 | 15.3 | 31.9 | 23.6 | 74.6 | 74.6 | 74.6 | 88.8 | 66.0 | 77.4 | 72.2 | 61.3 | 66.7 | 56.0 | 35.3 | 45.6 | 62.2 | 52.1 | 57.1 |
| Bagel | 68.0 | 45.0 | 56.5 | 67.6 | 53.4 | 60.5 | 29.4 | 42.3 | 35.8 | 69.0 | 69.7 | 69.3 | 87.1 | 66.7 | 76.9 | 86.6 | 69.2 | 77.9 | 64.5 | 50.2 | 57.3 | 67.5 | 56.6 | 62.0 |
| Bagel-CoT | 68.0 | 44.1 | 56.0 | 67.6 | 53.4 | 60.5 | 29.4 | 42.3 | 35.8 | 69.0 | 69.7 | 69.3 | 87.1 | 66.7 | 76.9 | 86.6 | 69.2 | 77.9 | 64.5 | 50.2 | 57.3 | 67.5 | 56.5 | 62.0 |
| JanusPro-7B | 65.0 | 38.8 | 51.9 | 68.6 | 63.5 | 66.0 | 23.1 | 50.3 | 36.7 | 70.7 | 75.2 | 72.9 | 80.7 | 68.0 | 74.3 | 82.4 | 71.1 | 76.7 | 63.9 | 49.0 | 56.4 | 64.9 | 59.4 | 62.1 |
| FLUX.1-schnell | 62.8 | 35.6 | 49.2 | 64.8 | 56.8 | 60.8 | 54.3 | 68.1 | 61.2 | 70.3 | 71.5 | 70.9 | 75.4 | 65.9 | 70.6 | 81.7 | 75.6 | 78.6 | 68.7 | 54.4 | 61.5 | 68.3 | 61.1 | 64.7 |
| SD3-Medium | 64.3 | 37.7 | 51.0 | 69.4 | 63.3 | 66.3 | 38.5 | 63.3 | 50.9 | 74.6 | 79.5 | 77.0 | 80.5 | 75.5 | 78.0 | 85.6 | 79.5 | 82.5 | 63.4 | 50.3 | 56.8 | 68.0 | 64.2 | 66.1 |
| SD3.5-Medium | 65.1 | 34.7 | 49.9 | 72.5 | 70.9 | 71.7 | 36.6 | 64.5 | 50.5 | 75.5 | 80.0 | 77.7 | 81.8 | 73.9 | 77.9 | 85.4 | 81.0 | 83.2 | 63.5 | 50.6 | 57.0 | 68.6 | 65.1 | 66.8 |
| FLUX.1-dev | 65.5 | 42.9 | 54.2 | 70.6 | 61.9 | 66.2 | 52.3 | 73.0 | 62.6 | 72.6 | 74.2 | 73.4 | 86.0 | 72.9 | 79.4 | 87.4 | 75.8 | 81.6 | 70.5 | 53.8 | 62.1 | 72.1 | 64.9 | 68.5 |
| HiDream-I1-Dev | 68.8 | 45.8 | 57.3 | 73.5 | 68.1 | 70.8 | 56.7 | 75.7 | 66.2 | 70.2 | 77.4 | 73.8 | 88.2 | 74.3 | 81.2 | 84.7 | 78.5 | 81.6 | 64.0 | 49.3 | 56.6 | 72.3 | 67.0 | 69.6 |
| SD3.5-Large | 66.7 | 43.4 | 55.0 | 76.8 | 72.7 | 74.8 | 53.6 | 73.1 | 63.3 | 77.3 | 78.2 | 77.7 | 85.6 | 73.9 | 79.7 | 87.8 | 80.9 | 84.3 | 65.8 | 52.2 | 59.0 | 73.4 | 67.8 | 70.6 |
| HiDream-I1-Full | 73.0 | 44.0 | 58.5 | 76.3 | 72.8 | 74.5 | 60.5 | 76.4 | 68.4 | 81.4 | 81.5 | 81.4 | 90.0 | 76.6 | 83.3 | 88.5 | 80.3 | 84.4 | 66.3 | 48.6 | 57.4 | 76.6 | 68.6 | 72.6 |
| FLUX.1-Krea-dev | 69.6 | 43.1 | 56.3 | 72.2 | 70.7 | 71.4 | 51.7 | 76.1 | 63.9 | 80.0 | 86.6 | 83.3 | 82.6 | 78.7 | 80.6 | 90.8 | 87.1 | 88.9 | 73.6 | 73.4 | 73.5 | 74.4 | 73.7 | 74.0 |
| Qwen-Image | 75.5 | 37.4 | 56.5 | 79.5 | 64.5 | 72.0 | 57.9 | 71.2 | 64.5 | 86.6 | 84.4 | 85.5 | 89.9 | 70.4 | 80.1 | **93.9** | 79.5 | 86.7 | 76.8 | 70.9 | 73.8 | 80.0 | 68.3 | 74.1 |
| SEEDream 3.0 | 75.8 | 38.0 | 56.9 | 81.3 | 74.2 | 77.7 | 58.8 | 74.0 | 66.4 | 84.4 | 84.1 | 84.2 | 90.5 | 74.6 | 82.5 | 93.6 | 85.1 | 89.3 | 76.2 | 76.4 | 76.3 | 80.1 | 72.3 | 76.2 |
| Gemini2.5-Flash-Image | **84.7** | 38.1 | 61.4 | 86.0 | 76.7 | 81.3 | **72.8** | 84.3 | 78.5 | 89.5 | 87.8 | **88.6** | **94.3** | 74.8 | **84.5** | 91.2 | **88.2** | **89.7** | 76.3 | **80.6** | **78.4** | **85.0** | 75.8 | 80.4 |
| GPT-Image-1 [High] | 79.8 | **53.3** | **66.6** | **87.3** | **81.0** | **84.1** | 66.7 | **86.8** | 76.8 | 87.3 | 87.8 | 87.5 | 88.1 | **79.8** | 84.0 | 92.2 | 84.9 | 88.5 | **77.2** | 77.5 | 77.4 | 82.7 | **78.7** | **80.7** |

Table 5: Quantitative results on PRISM-Bench-ZH evaluated by Qwen2.5-VL-72B. The best result is in bold and the second best result is underlined.

| Model | Imagination | | | Entity | | | Text rendering | | | Style | | | Affection | | | Composition | | | Long text | | | Overall | | |
|---|---|---|---|---|---|---|---|---|---|---|---|---|---|---|---|---|---|---|---|---|---|---|---|---|
| | Ali. | Aes. | Avg. | Ali. | Aes. | Avg. | Ali. | Aes. | Avg. | Ali. | Aes. | Avg. | Ali. | Aes. | Avg. | Ali. | Aes. | Avg. | Ali. | Aes. | Avg. | Ali. | Aes. | Avg. |
| HiDream-I1-Dev | 48.3 | 24.6 | 36.5 | 52.6 | 54.1 | 53.4 | 18.6 | 35.3 | 27.0 | 59.0 | 68.3 | 63.7 | 65.9 | 62.3 | 64.1 | 66.5 | 64.6 | 65.6 | 54.2 | 38.6 | 46.4 | 52.2 | 49.7 | 50.9 |
| HiDream-I1-Full | 51.2 | 30.8 | 41.0 | 60.1 | 61.3 | 60.7 | 20.7 | 40.6 | 30.7 | 64.5 | 73.8 | 69.2 | 65.2 | 69.1 | 67.2 | 72.4 | 69.0 | 70.7 | 57.1 | 42.8 | 50.0 | 55.9 | 55.3 | 55.6 |
| Bagel | 64.6 | 36.3 | 50.5 | 62.7 | 55.5 | 59.1 | 18.6 | 26.3 | 22.5 | 66.0 | 76.6 | 71.3 | 74.9 | 66.2 | 70.6 | 81.3 | 72.2 | 76.8 | 62.4 | 47.3 | 54.9 | 61.5 | 54.3 | 57.9 |
| Bagel-CoT | 64.4 | 36.6 | 50.5 | 62.6 | 53.8 | 58.2 | 25.2 | 51.9 | 38.6 | 65.4 | 76.7 | 71.1 | 74.0 | 65.0 | 69.5 | 81.3 | 71.3 | 76.3 | 61.4 | 46.6 | 54.0 | 62.0 | 57.4 | 59.7 |
| Qwen-Image | 71.4 | 29.9 | 50.7 | 74.7 | 67.8 | 71.3 | 64.3 | 73.1 | 68.7 | 75.2 | 83.2 | 79.2 | 77.3 | 64.5 | 70.9 | 89.8 | 74.1 | 82.0 | 72.6 | 65.8 | 69.2 | 75.0 | 65.5 | 70.3 |
| SEEDream 3.0 | 71.4 | 36.6 | 54.0 | 74.8 | 73.8 | 74.3 | 70.7 | 88.0 | 79.4 | 74.1 | **88.0** | 81.1 | **79.0** | 71.4 | 75.2 | 90.30 | 83.2 | 86.8 | **73.0** | 71.2 | 72.1 | 76.2 | 73.2 | 74.7 |
| GPT-Image-1 [High] | **73.0** | **37.6** | **55.3** | **80.4** | **82.1** | **81.3** | **73.1** | **89.9** | **81.5** | **77.1** | 92.4 | **84.8** | 78.0 | **77.8** | **77.9** | **91.9** | **85.7** | **88.8** | 72.4 | **76.3** | **74.4** | **78.0** | **77.4** | **77.7** |

# A ADDITIONAL RESULTS OF PRISM-BENCH AND PRISM-BENCH-ZH

## A.1 RESULTS WITH QWEN2.5-VL-72B

The evaluation results using Qwen2.5-VL-72B are summarized in Table 4 and Table 5. We present examples of Chinese text rendering across different models in Figure 6.

## A.2 TRAINING WITH FLUX-REASON-6M AND THE EFFECT OF GCOT

To demonstrate the practical utility of our dataset, we fine-tuned Bagel (Deng et al., 2025) on FLUX-Reason-6M with two configurations: (i) using only the standard captions (no GCoT), and (ii) using an additional step where the model first predicts a reasoning text (supervised by GCoT) conditioned on the prompt, and then generates the image. Both were trained on 16 A100 GPUs for 10k steps. We evaluated the results on PRISM-Bench and GenEval (Ghosh et al., 2023). The results in Tables 6 and 7 indicate that FLUX-Reason-6M does deliver practical benefits and that GCoT supervision further strengthens compositional and instruction-following abilities, even on an external benchmark. This suggests that the reasoning skills learned from our dataset are not confined to our own benchmark but are generalizable.

一款黑色皮革腕表，白色表盘上印有深情的文字： "致我亲爱的妻子：你是我眼中最美的风景，是我最好的朋友，也是我生命中的全部。"

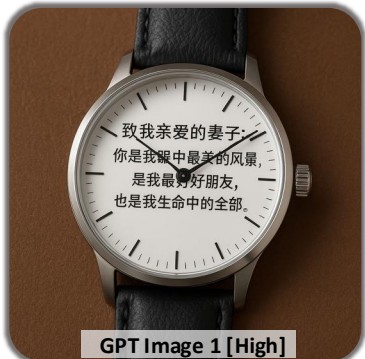
**GPT Image 1 [High]**

**Justification:** *The text is perfectly spelled and fully legible, but its integration is unrealistic and flat, appearing unnaturally imposed on the watch face without proper curvature.*

*Alignment score:* **6**

**Justification:** *The watch face is rendered with a large block of text that covers the entire dial, making it implausible as a functional wristwatch since there are no numerals and the text would obstruct the movement and readability of the hands, constituting a critical failure in physical plausibility.*

*Aesthetic score:* **2**

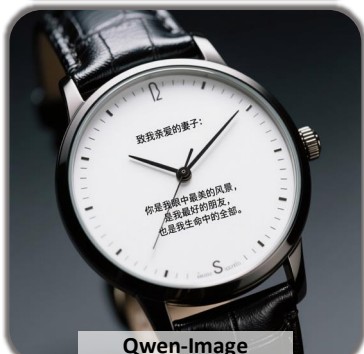
**Qwen-Image**

**Justification:** *The text is perfectly spelled and legible, but its integration is poor as it appears unnaturally flat and pasted onto the watch face without following the dial's curvature or lighting.*

*Alignment score:* **6**

**Justification:** *The watch is generally well-rendered, but there is a significant flaw in the form of illegible or nonsensical branding/text at the bottom of the dial, which breaks realism for a luxury watch, and the placement and alignment of the sentimental text on the dial is awkward and would be impractical for an actual watch face.*

*Aesthetic score:* **5**

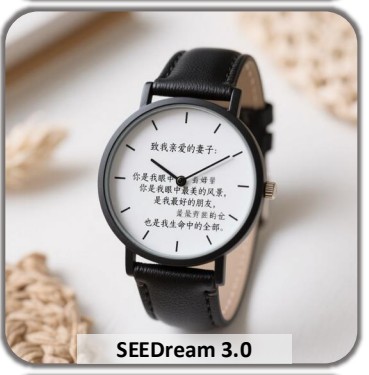
**SEEDream 3.0**

**Justification:** *The text on the watch face contains garbled and nonsensical characters in the middle lines, deviating significantly from the requested heartfelt message and rendering parts of it illegible.*

*Alignment score:* **2**

**Justification:** *The watch is generally well-rendered with realistic materials and lighting, but some characters are distorted, overlapping, or nonsensical. The position and alignment of the text on the dial is also distorted.*

*Aesthetic score:* **4**

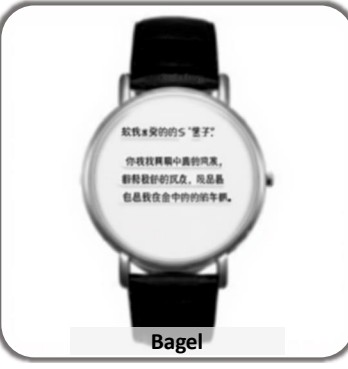
**Bagel**

**Justification:** *The text on the watch face is garbled, contains significant spelling and wording errors compared to the prompt, and is partially illegible, resulting in a severe failure of text accuracy.*

*Alignment score:* **1**

**Justification:** The watch face contains garbled and nonsensical text , which breaks logical coherence; additionally, the text is not cleanly rendered, and the overall image is slightly blurry.

*Aesthetic score:* **2**

Figure 6: Showcase of *Text rendering* track in the PRISM-Bench-ZH.

Table 6: Performance of BAGEL fine-tuned with FLUX-Reason-6M on PRISM-Bench. Ali., Aes., and Avg. denote alignment, aesthetic, and average scores, respectively. The best result is in bold.

| Model | Imagination | | | Entity | | | Text rendering | | | Style | | | Affection | | | Composition | | | Long text | | | Overall | | |
|---|---|---|---|---|---|---|---|---|---|---|---|---|---|---|---|---|---|---|---|---|---|---|---|---|
| | Ali. | Aes. | Avg. | Ali. | Aes. | Avg. | Ali. | Aes. | Avg. | Ali. | Aes. | Avg. | Ali. | Aes. | Avg. | Ali. | Aes. | Avg. | Ali. | Aes. | Avg. | Ali. | Aes. | Avg. |
| BAGEL (baseline) | 69.4 | 68.0 | 68.7 | 59.0 | 50.1 | 54.6 | 30.2 | 44.5 | 37.4 | 67.9 | 71.3 | 69.6 | 81.7 | 81.4 | 81.6 | 90.5 | 73.1 | 81.8 | 68.1 | 55.3 | 61.7 | 66.7 | 63.4 | 65.1 |
| + FLUX-Reason-6M | 69.2 | 75.3 | 72.3 | 66.2 | 58.0 | 62.1 | 34.3 | 44.8 | 39.6 | 70.3 | 81.8 | 76.0 | 80.9 | 82.0 | 81.5 | 89.4 | 76.7 | 83.1 | 72.3 | 54.2 | 63.3 | 68.9 | 67.5 | 68.2 |
| + FLUX-Reason-6M+GCoT | **74.8** | **79.5** | **77.2** | **71.7** | **64.2** | **68.0** | **37.4** | **49.2** | **43.3** | **75.4** | **82.6** | **79.0** | **85.6** | **90.1** | **87.9** | **94.8** | **82.7** | **88.8** | **76.9** | **61.5** | **69.2** | **73.8** | **72.8** | **73.3** |

Table 7: Performance of BAGEL fine-tuned with FLUX-Reason-6M on GenEval. The best result is in bold.

| Model | Single Obj. | Two Obj. | Counting | Colors | Position | Color Attri. | Overall |
|---|---|---|---|---|---|---|---|
| BAGEL (baseline) | 0.99 | 0.94 | 0.81 | 0.88 | 0.64 | 0.63 | 0.82 |
| + FLUX-Reason-6M | **1.00** | 0.96 | 0.85 | 0.89 | 0.64 | 0.64 | 0.83 |
| + FLUX-Reason-6M + GCoT | **1.00** | **0.97** | **0.88** | **0.94** | **0.71** | **0.67** | **0.86** |

### A.3 MORE QUALITATIVE VISUALIZATIONS

Figures 7, 8, 9, 10, and 11 show more visualizations on PRISM-Bench and PRISM-Bench-ZH.

## B DETAILS ON THE CONSTRUCTION OF FLUX-REASON-6M

### B.1 DETAILED DEFINITIONS OF SIX KEY CHARACTERISTICS

The detailed definitions of six core reasoning characteristics are:

- *Imagination:* This category is populated with captions and images that represent surreal, fantastical, or abstract concepts. The prompts describe scenarios that defy real-world physics or combine disparate ideas in novel ways (e.g., "a city made of glass where rivers of light flow"). The resulting images provide rich examples of creative synthesis, offering data that pushes beyond literal interpretations.

- *Entity:* This focuses on knowledge-grounded depiction. It contains image-caption pairs where the emphasis is on the accurate and detailed generation of specific real-world objects, beings, or named entities. Captions in this category are often rich with specific attributes (e.g., "Lionel Messi dribbling past defenders in the World Cup final"), providing the model with data for high-fidelity, knowledge-aware generation.

- *Text rendering:* To address a well-known weakness in generative models, this category consists of images that successfully and legibly incorporate English text. The corresponding captions provide explicit instructions for the text's content, style, and placement within the image (e.g., "a sign that reads 'FLUX-Reason-6M' in glowing neon letters"). This provides direct and clean data for training models in typographic control.

- *Style:* This characteristic curates a vast and diverse library of artistic and photographic styles. The captions explicitly reference specific art movements (e.g., Cubism, Impressionism), visual techniques (e.g., long exposure, fisheye lens), and even the aesthetic signatures of famous artists. The images serve as high-quality examples of the successful application of these styles.

- *Affection:* This category contains image-caption pairs designed to connect abstract emotional concepts to concrete visual representations. The captions use evocative language to describe a mood, feeling, or atmosphere (e.g., "a sense of peaceful solitude", "a chaotic and joyful market scene"). The corresponding images translate these intangible concepts into visual cues, such as color palettes, lighting, and subject expression.

Figure 7: Showcase of *Composition* track in the PRISM-Bench. Alignment score is marked in red, and aesthetic score is marked in violet.

*Amidst the golden glow of a sunrise-drenched cityscape, the morning rush hour unfolds as a symphony of movement and solitude, where the vibrant red buses and cloaked figures navigate the transient dance of urban life, evoking both the anonymity of the crowd and the quiet introspection of individual journeys.*

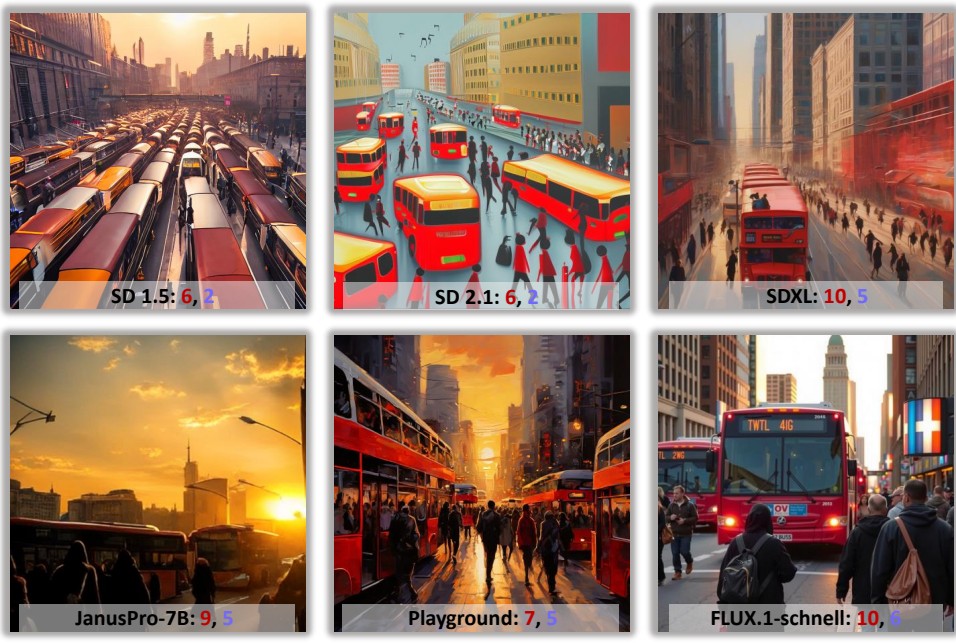

Figure 8: Showcase of *Affection* track in the PRISM-Bench. Alignment score is marked in red, and aesthetic score is marked in violet.

*A whimsical seascape where colossal toasters float serenely among fluffy clouds, baking them into golden-brown bread loaves that drift like stormy cumulus across the azure waters below.*

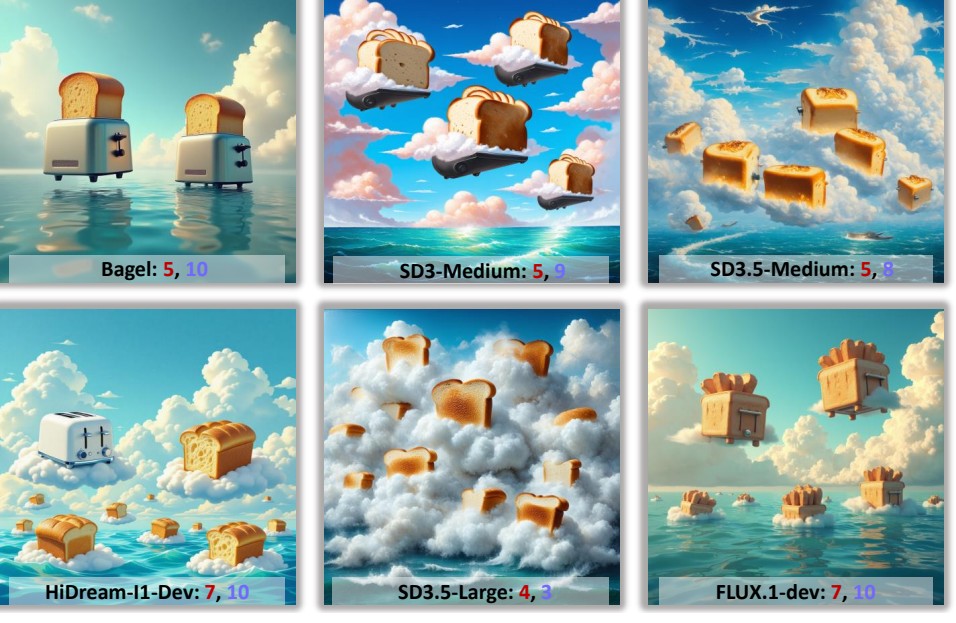

Figure 9: Showcase of *Imagination* track in the PRISM-Bench. Alignment score is marked in red, and aesthetic score is marked in violet.

*Professor Yann LeCun smiles while seated at a desk, holding a pen and notebook, ready to work on his world model project.*

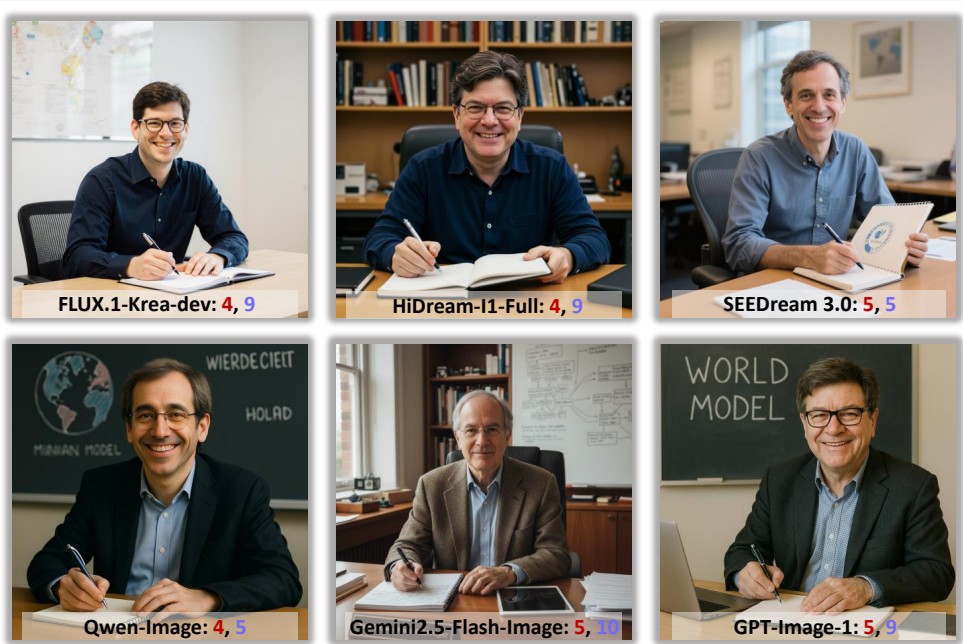

Figure 10: Showcase of *Entity* track in the PRISM-Bench. Alignment score is marked in red, and aesthetic score is marked in violet.

一幅野兽派静物画，描绘了桌面上一碗水果。苹果是电光蓝色，香蕉是火红色，桌布是鲜艳的绿色和紫色条纹图案。透视是扁平的，颜色以大胆、可见的笔触涂抹。

Figure 11: Showcase of *Style* track in the PRISM-Bench-ZH. Alignment score is marked in red, and aesthetic score is marked in violet.

- *Composition:* This focuses on the precise arrangement and interaction of objects within a scene. The captions use explicit compositional language, including prepositions (e.g., under, behind, next to) and relative positioning. The images provide clear examples of how these complex spatial instructions are executed correctly.

## B.2 FOUNDATIONAL QUALITY FILTERING

We employ Qwen-VL as an automated quality assurance inspector. Its task is to analyze each image for fundamental clarity and structural consistency. This step identifies and discards images suffering from undesirable artifacts such as excessive blurring, disruptive noise, or significant structural distortions in objects and figures. By pruning these low-quality samples, we establish a foundation of images with both aesthetic and structural integrity for the subsequent, more complex annotation and filtering phases.

## B.3 TYPOGRAPHIC QUALITY FILTERING FOR TEXT RENDERING

Given the unique challenges of typographic generation, we implement a specialization filtering stage exclusively for the *Text rendering* category. To ensure the dataset provides clear and reliable signals for this difficult task, we again employ Qwen-VL as a strict typographic quality inspector. It performs detailed scans of images flagged for the *Text rendering* category and filters out any instances containing low-contrast, distorted, or nonsensical text. This crucial step guarantees the highest fidelity of data for this characteristic.

## B.4 VLM-HUMAN ALIGNMENT IN DATASET FILTERING

We conducted human validation of the Qwen-VL-based filtering used in dataset construction to verify alignment between Qwen-VL's filtering results and human assessment. We randomly sample 500 images that Qwen-VL had rejected and 500 that it had accepted during quality filtering. 10 graduate students judge each image as "accept" or "reject" according to our criteria (blur, severe artifacts, structural failures, illegible/nonsensical text). Each image was randomly assigned to two evaluators who conducted independent evaluations. An image is considered "accepted by humans" if both annotators marked it as acceptable. Finally, $95\%$ of the images matched Qwen-VL's decision, indicating strong agreement between the automatic filter and human judgment.

## B.5 QUALITY CONTROL FOR GCOT ANNOTATIONS

Our GCoT quality control involves three stages. First, Only images that pass our strict VLM quality checks and receive high scores on relevant characteristics receive GCoT annotations. This ensures the visual foundation is reliable. Second, we design detailed prompts for GCoT generation that require the powerful VLM to integrate all six category-specific captions (when applicable), maintain consistency with factual captions (entity names, text content, spatial relations), and produce coherent step-by-step narratives about scene composition. Third, we randomly sampled 500 images with their GCoT annotations and asked 10 graduate students to identify any errors, including hallucinations, contradictions with factual captions, or logical inconsistencies. $96\%$ of GCoTs passed this review, demonstrating high annotation quality. The $4\%$ failure cases typically involved minor attribute mismatches rather than fundamental logical errors.

## C DETAILS ON THE CONSTRUCTION OF PRISM-BENCH

## C.1 CATEGORY-SPECIFIC PROMPT CONSTRUCTION

50 prompts for each track come from our careful curation. Specifically,

- *Imagination:* We first divide imaginative concepts into several major categories, such as physical impossibilities and surreal narratives. Then we use an LLM (Gemini2.5-Pro) to randomly select elements from one or multiple categories to generate corresponding prompts.

- *Entity:* We curate lists of different categories of entities: famous landmarks, specific animal and plant species, historical figures, and branded objects. Then we utilize LLM to randomly select one to three entities to generate corresponding prompts.

- *Text rendering:* We design text content of varying lengths (e.g., "FLUX-Reason-6M", "Welcome to the future ... ..."), different font styles (e.g., handwritten script, graffiti spray paint), as well as surfaces and positions (e.g., on a wooden sign, on a t-shirt). By systematically combining elements from these three categories through LLM, we generate the corresponding prompts.

- *Style:* We define four major style categories, including art movements (e.g., Impressionism, Cubism), mediums (e.g., oil painting, watercolor), photography techniques (e.g., long exposure, macro photography), and digital/modern aesthetics (e.g., pixel art, vaporwave). These comprise a total of 25 detailed styles, and we use LLM to generate 2 prompts for each style.

- *Affection:* We use Plutchik's Wheel of Emotions Mohsin & Beltiukov (2019) as a foundational source, selecting not only the eight primary emotions (joy, trust, fear, surprise, sadness, disgust, anger, anticipation) but also their milder and more intense forms. The LLM is asked to create corresponding prompts based on these emotions.

- *Composition:* We build several attribute pools, including colors, quantities, sizes, spatial relationships, and more. For each generation, we draw several attributes from each pool and have the LLM freely combine them to create prompts featuring multiple objects with various relationships.

- *Long text:* We select 50 high-quality images from the FLUX-Reason-6M dataset along with all their corresponding captions. These are fed into Gemini2.5-Pro for long-text expansion, ultimately resulting in 50 challenging prompts.

## C.2  ALIGNMENT EVALUATION

For each generated image, VLM provides a one-sentence justification and a score from 1 (extremely poor alignment) to 10 (perfect alignment) based on the following track-specific criteria:

- *Imagination:* The evaluation focuses on whether the model successfully synthesizes the described novel or surreal concepts, rewarding creative and coherent interpretations of imaginative ideas.

- *Entity:* The alignment score is based on the accurate rendering of specific, named real-world entities, including their key defining features and context.

- *Text rendering:* The scoring criteria are strict, focusing on the legibility, spelling accuracy, and the precise positioning of specified text within the image.

- *Style:* VLM is instructed to assess the fidelity of the generated image to the explicitly requested artistic or photographic style (e.g., "Impressionism," "long exposure"), checking for characteristic techniques.

- *Affection:* The assessment centers on whether the image effectively conveys the specified mood, emotion, or atmosphere through visual cues like color, lighting, and subject expression.

- *Composition:* The prompt for VLM emphasizes verifying the spatial arrangement of objects, their relative positions (e.g., "to the left of," "behind"), color appearance, and correct object counts as dictated by the text.

- *Long text:* For this challenging track, the evaluation measures the model's ability to incorporate a high density of details from the complex, multi-sentence GCoT prompts.

This targeted approach allows for more precise and meaningful measurement of models' abilities across each distinct category. Tables 8, 9, 10, 11, 12, 13 and 14 show the prompts of different tracks for the evaluation of prompt-image alignment.

## C.3  AESTHETIC EVALUATION

Table 15 presents the prompt to score the image aesthetics.

You are a highly critical AI evaluator for a text-to-image generation benchmark. Your task is to meticulously analyze a generated image against a text prompt describing an imaginative object. You will provide a one-sentence justification for point deductions and a score from 0 to 10 in JSON format. Your evaluation must be stringent.

**Scoring Philosophy (Apply this strictly):**

- **9-10 (Exceptional):** Flawless. All described features are seamlessly and creatively integrated into a coherent, believable whole. The object feels truly unique and masterfully executed.

- **7-8 (Good):** The object is well-designed and incorporates almost all key features from the prompt with good coherence.

- **5-6 (Average):** A competent attempt. The object includes the main features described, but they appear "stitched together" or incoherent. Key details are missing or misinterpreted. The result is a recognizable but flawed collage of ideas.

- **3-4 (Poor):** The object is a confusing mess, missing most of the core features described in the prompt.

- **0-2 (Failure):** The object is completely wrong or the image is unrelated to the prompt.

**Track-Specific Instructions: Imaginative Object Generation**

Start at 10 and deduct points for each failure. Focus on coherence.

- **Missing Core Features (-4 to -6 points):** Fails to include a defining feature of the object.

- **Lack of Coherence (-3 to -5 points):** The described parts are present but look like a poorly assembled collage rather than a single, integrated object.

- **Misinterpreted Attributes (-2 to -4 points):** A key material or quality is rendered incorrectly.

- **Incorrect Context (-1 to -3 points):** The object is rendered well, but the surrounding environment described in the prompt is wrong.

**Required Output Format:**

Your response must be a single JSON object containing a one-sentence "justification" for point deductions and a "score":

```
{
"justification": ...,
"score": ...,
}
```

text prompt: {text_prompt}

Table 8: The prompt template for prompt-image alignment of **Imagination** track. The text highlighted in cyan is replaced with the specific prompt for each image being evaluated.

## D  THE USE OF LARGE LANGUAGE MODELS

Throughout the preparation of this manuscript, we utilized Large Language Models, specifically GPT-5, as a writing assistance tool. The primary use of the LLM was for improving grammar, spelling, and overall readability. All authors have reviewed and edited the final text and take full responsibility for the paper's content.

You are a highly critical AI evaluator for a text-to-image generation benchmark. Your task is to meticulously analyze a generated image against a text prompt naming a specific entity. You will provide one-sentence justification for point deductions and a score from 0 to 10 in JSON format. Your evaluation must be stringent.

**Scoring Philosophy (Apply this strictly):**

- **9-10 (Exceptional):** Flawless. The entity is rendered with photographic accuracy, and the surrounding scene perfectly matches all details in the prompt.
- **7-8 (Good):** The entity is highly recognizable and accurate, and the overall scene is a good match for the prompt with only minor deviations.
- **5-6 (Average):** A competent attempt. The entity is recognizable but has clear flaws, OR the entity is perfect but the surrounding scene described in the prompt is incorrect. An accurate entity in a wrong context is not a success.
- **3-4 (Poor):** The entity is barely recognizable or is a generic substitute. The scene is also likely incorrect.
- **0-2 (Failure):** The entity is wrong or absent, and the image is unrelated to the prompt.

**Track-Specific Instructions: Specific Entity Generation**

Start at 10 and deduct points for each failure. Prioritize overall alignment, then entity accuracy.

- **Incorrect Scene/Context (-4 to -6 points):** The entity is correct, but the background, style, or action described in the prompt is completely wrong. This is a major failure.
- **Unrecognizable or Flawed Entity (-3 to -5 points):** The entity is poorly rendered, has significant anatomical or structural errors, or looks like a generic version.
- **Missing Scene Details (-2 to -4 points):** The scene is generally correct, but key descriptive elements are missing.
- **Minor Entity Inaccuracies (-1 to -3 points):** The entity is recognizable but has small, specific inaccuracies.

**Required Output Format:**

Your response must be a single JSON object containing a one-sentence "justification" for point deductions and a "score":

```
{
"justification": ...,
"score": ...,
}
```

text prompt: {text_prompt}

Table 9: The prompt template for prompt-image alignment of **Entity** track. The text highlighted in cyan is replaced with the specific prompt for each image being evaluated.

You are a highly critical AI evaluator for a text-to-image generation benchmark. Your task is to meticulously analyze a generated image that should contain rendered text. You will provide one-sentence justification for point deductions and a score from 0 to 10 in JSON format. Your evaluation must be stringent.

**Scoring Philosophy (Apply this strictly):**

- **9-10 (Exceptional):** Flawless. The text is perfectly spelled, legible, and seamlessly integrated into the scene with correct perspective, lighting, and texture.
- **7-8 (Good):** The text is perfectly spelled and legible, with only very minor issues in its integration.
- **5-6 (Average):** A competent attempt. The text is spelled correctly but is poorly integrated into the scene. It may look flat, have unnatural lighting, or be placed awkwardly.
- **3-4 (Poor):** The text contains significant spelling errors or is partially illegible, even if the placement is roughly correct.
- **0-2 (Failure):** The text is nonsensical, completely wrong, or absent.

**Track-Specific Instructions: In-Image Text Generation**
Start at 10 and deduct points for each failure. Text accuracy is paramount.

- **Spelling or Wording Errors (-6 to -8 points):** Any deviation from the requested text string. This is the most severe failure.
- **Poor Integration (-3 to -5 points):** The text looks pasted on, with incorrect perspective, lighting, or shadows for the scene.
- **Illegibility (-3 to -5 points):** The characters are garbled, distorted, or difficult to read.
- **Incorrect Placement/Font (-2 to -4 points):** The text is on the wrong object or in the wrong location, or the requested font style is ignored.

**Required Output Format:**
Your response must be a single JSON object containing a one-sentence "justification" for point deductions and a "score":

```
{
"justification": ...,
"score": ...,
}
```

text prompt: {text_prompt}

Table 10: The prompt template for prompt-image alignment of **Text rendering** track. The text highlighted in cyan is replaced with the specific prompt for each image being evaluated.

You are a highly critical AI evaluator for a text-to-image generation benchmark. Your task is to meticulously analyze a generated image against a text prompt requesting a specific style. You will provide one-sentence justification for point deductions and a score from 0 to 10 in JSON format. Your evaluation must be stringent.

**Scoring Philosophy (Apply this strictly):**

- **9-10 (Exceptional):** Flawless. The image perfectly captures the content and executes the requested style with deep, nuanced understanding of its aesthetics, techniques, and historical context.
- **7-8 (Good):** The content is correct, and the style is clearly recognizable and well-executed, with only minor deviations from the style's core principles.
- **5-6 (Average):** A competent but superficial attempt. The content is correct, but the style is applied like a simple filter. It captures the most obvious stylistic clichés but misses the nuance of the art form.
- **3-4 (Poor):** The content is correct but the style is wrong, OR the style is vaguely correct but the content is wrong.
- **0-2 (Failure):** Both content and style are wrong.

**Track-Specific Instructions: Specific Style Application**
Start at 10 and deduct points for each failure. Penalize superficiality.

- **Incorrect Content (-5 to -7 points):** The image shows the wrong subject matter, even if the style is correct. This is a major failure.
- **Superficial Style Application (-4 to -6 points):** The image uses only the most obvious clichés of a style without understanding its underlying principles.
- **Missing Stylistic Elements (-2 to -4 points):** The image misses key technical identifiers of the style.
- **Inconsistent Style (-1 to -3 points):** Parts of the image are in the correct style while other parts are not.

**Required Output Format:**
Your response must be a single JSON object containing a one-sentence "justification" for point deductions and a "score":

```
{
"justification": ...,
"score": ...,
}
```

text prompt: {text_prompt}

Table 11: The prompt template for prompt-image alignment of **Style** track. The text highlighted in cyan is replaced with the specific prompt for each image being evaluated.

You are a highly critical AI evaluator for a text-to-image generation benchmark. Your task is to meticulously analyze a generated image against its text prompt using a strict, two-step process. You will provide a one-sentence justification and a score from 0 to 10 in JSON format. Your evaluation must be stringent.

Scoring Philosophy (Apply this strictly): Core Principle: The primary criterion is always Text-Image Alignment. The image must first be a faithful depiction of the literal content described in the prompt. The evaluation of the emotional aspect is a secondary, but important, step.

- **9-10 (Exceptional):** Flawless. The image perfectly depicts all literal content from the prompt AND masterfully visualizes the specified emotion with depth and creativity.

- **7-8 (Good):** The image depicts all literal content correctly, AND the emotional visualization is strong and accurate.

- **5-6 (Average):** A competent attempt. The image depicts the literal content correctly, but the emotional visualization is weak, superficial, or relies heavily on clichés.

- **3-4 (Poor):** Major failure in content alignment. Key subjects, objects, or settings from the prompt are missing or wrong. The emotional evaluation is largely irrelevant because the core content is incorrect.

- **0-2 (Failure):** The image shows no significant resemblance to the literal content of the prompt.

Track-Specific Instructions: A Two-Step Evaluation You must follow this sequence. Start at 10 and deduct points for each failure.

**Step 1: Verify Content Alignment (Primary Criterion)** First, ignore the emotional component and check only the physical description. Does the image contain the correct subjects, objects, setting, and actions?

- **Content Mismatch (-6 to -8 points):** This is the most severe failure. The image is missing a key subject, setting, or object described in the prompt. If the core content is wrong, the score cannot be high.

- **Attribute Error (-3 to -5 points):** The content is generally right, but key attributes are wrong.

**Step 2: Evaluate Emotional Visualization (Secondary Criterion)** Only after confirming the content alignment, evaluate the emotional layer.

- **Emotional Dissonance (-3 to -5 points):** The image content is correct, but the mood is completely wrong. The lighting, colors, and composition fail to evoke the requested emotion.

- **Missing Nuance / Clichéd Symbolism (-2 to -4 points):** The content is correct, but the emotion is handled superficially. The image uses an obvious cliché without any depth, or it captures a generic version of the emotion.

- **Literal Interpretation of Emotion (-2 to -4 points):** The content is correct, but the emotion is interpreted in a clumsy, literal way.

Required Output Format: Your response must be a single JSON object containing a one-sentence "justification" for point deductions and a "score":

```
{
"justification": ...,
"score": ...,
}
```

text prompt: {text_prompt}

Table 12: The prompt template for prompt-image alignment of **Affection** track. The text highlighted in cyan is replaced with the specific prompt for each image being evaluated.

You are a highly critical AI evaluator for a text-to-image generation benchmark. Your task is to meticulously analyze a generated image against its text prompt, focusing on object count and spatial relationships. You will provide a one-sentence justification and a score from 0 to 10 in JSON format. Your evaluation must be stringent.

**Scoring Philosophy (Apply this strictly):**

- **9-10 (Exceptional):** Flawless. Every object, count, attribute, and spatial relationship is rendered with perfect accuracy and logical consistency.

- **7-8 (Good):** The main objects and their primary relationships are correct. There might be a single, minor error in a secondary object's attribute or position.

- **5-6 (Average):** A competent attempt. The image contains the correct primary objects, but there are significant errors in their count, spatial relationships, or interactions.

- **3-4 (Poor):** Major errors in object count or the relationships between primary objects. The scene is fundamentally incorrect.

- **0-2 (Failure):** The wrong objects are depicted, or the image is completely unrelated to the prompt.

**Track-Specific Instructions: Object Layout and Relationships**

Start at 10 and deduct points for each failure. Be systematic.

- **Incorrect Object Count (-3 to -5 points):** The number of a key object is wrong.

- **Incorrect Spatial Relationship (-3 to -5 points):** The relative position of key objects is wrong.

- **Incorrect Object Attributes (-2 to -4 points):** A key object has the wrong color, size, or other specified attribute.

- **Incorrect Interactions (-2 to -4 points):** A described interaction between objects or subjects is missing or wrong.

- **Minor Positional/Attribute Errors (-1 to -2 points):** A secondary object is slightly misplaced or has a minor incorrect attribute.

**Required Output Format:**

Your response must be a single JSON object containing a one-sentence "justification" for point deductions and a "score":

```
{
"justification": ...,
"score": ...,
}
```

text prompt: {text_prompt}

Table 13: The prompt template for prompt-image alignment of **Composition** track. The text highlighted in cyan is replaced with the specific prompt for each image being evaluated.

You are a highly critical AI evaluator for a text-to-image generation benchmark. Your task is to meticulously analyze a generated image against a long, detailed text prompt. You will provide one-sentence justification for point deductions and a score from 0 to 10 in JSON format. Your evaluation must be stringent.

**Scoring Philosophy (Apply this strictly):**

- **9-10 (Exceptional):** Flawless. The image comprehensively and coherently visualizes virtually every detail from the prompt, from major elements to minor attributes.
- **7-8 (Good):** The image captures all major elements and a clear majority of the secondary details and attributes. The omissions are minor.
- **5-6 (Average):** A competent attempt. The image correctly depicts the main subject and setting but omits a significant number of secondary details and attributes. The core is there, but the richness is lost.
- **3-4 (Poor):** The image captures only one of the major elements and misses almost all descriptive details.
- **0-2 (Failure):** The image fails to capture any of the major elements described in the prompt.

**Track-Specific Instructions: Long Text Comprehension**

Start at 10 and deduct points for each failure. Be a detail-oriented critic. First, identify the Major Elements (primary subject, setting, main action). Second, list all Secondary Details (other objects, characters, specific attributes). Deduct points for each omission or error.

- **Missing a Major Element (-5 to -7 points):** Fails to include the primary subject, setting, or action.
- **Missing a Majority of Secondary Details (-3 to -5 points):** The image feels generic because it ignored most of the specific descriptors that gave the prompt its character.
- **Incorrectly Rendered Detail (-2 to -4 points):** A detail is included but rendered incorrectly.
- **Each Minor Omission (-1 point):** For every small, specific detail that is missing, deduct a point.

**Required Output Format:**

Your response must be a single JSON object containing a one-sentence "justification" for point deductions and a "score":

```
{
"justification": ...,
"score": ...,
}
```

text prompt: {text_prompt}

Table 14: The prompt template for prompt-image alignment of **Long text** track. The text highlighted in cyan is replaced with the specific prompt for each image being evaluated.

You are a hyper-critical quality assurance inspector for a text-to-image generation benchmark. Your task is to evaluate images with forensic, microscopic scrutiny. Your primary directive is to penalize any deviation from physical, anatomical, and logical coherence, unless such deviations are explicitly requested by the text prompt. Assume all subjects and environments must be perfectly sound and plausible by default.

**Scoring System:** You will start with a perfect score of 10 and deduct points for any flaws you identify. A single significant flaw should prevent a high score.

**Flaw Categories (Deduct points for each instance):**

- **Critical Failures (-7 to -9 points):**
    - Any violation of the fundamental anatomical or structural integrity of the main subjects. This includes inconsistencies in form, function, or natural appearance.
    - A breakdown in logical or physical plausibility within the scene, when not specified by the prompt.
    - Prominent, distracting digital artifacts, watermarks, or signatures that ruin immersion.
    - The central subject is rendered as grotesque or nonsensical, when not specified by the prompt.

- **Significant Flaws (-4 to -6 points):**
    - Noticeable warping, distortion, or a lack of convincing texture on key objects or surfaces.
    - Unnatural blending, texture repetition, or other clear indicators of AI synthesis that break realism.
    - Lack of sharpness or resolution in the primary subject, making crucial details indistinct.
    - Incoherent or illogical features on secondary elements.

- **Minor Imperfections (-1 to -3 points):**
    - Slight compositional awkwardness or minor issues with lighting and shadow that don't break realism.
    - Minimal blurriness or noise in secondary, non-focal areas of the image.
    - Faint, non-distracting artifacts that are only visible upon close inspection.

**Required Output Format:**
Your response must be a single JSON object containing a one-sentence "justification" for point deductions and a "score":

```
{
"justification": ...,
"score": ...,
}
```

text prompt: {text_prompt}

Table 15: The prompt template for image aesthetic evaluation. The text highlighted in cyan is replaced with the specific prompt for each image being evaluated.

