# OpenReview forum: "FLUX-Reason-6M & PRISM-Bench: A Million-Scale Text-to-Image Reasoning Dataset and Comprehensive Benchmark"
_ICLR.cc/2026/Conference — ICLR 2026 Poster_

### Official Review · Reviewer_oMcM · 2025-10-26

**Soundness:** 3
**Presentation:** 2
**Contribution:** 3
**Rating:** 4
**Confidence:** 4

**Summary:**

This paper introduces FLUX-Reason-6M and PRISM-Bench. FLUX-Reason-6M is a 6-million-scale synthesized dataset curated by Flux, with the first multi-dimensional, million-scale generation chain-of-thought annotations.  PRISM-Bench serves as a comprehensive and discriminative benchmark with 7 independent tracks that evaluate Imagination, Entity, Text rendering, Style, Affection, and Composition. By evaluating 19 leading models, the work identifies critical performance gaps that require improvement.

**Strengths:**

Unlike many datasets that are simple collections of image-caption pairs, FLUX-Reason-6M is specifically designed to teach complex reasoning capabilities to T2I models. It moves beyond merely describing what is in an image to also explaining why it is composed in a particular way.

A major innovation is the introduction of GCoT, which provides detailed breakdowns of image generation steps. These explicit reasoning paths deconstruct the semantic and compositional logic of the final image, offering powerful intermediate supervisory signals for training models to understand underlying structures and artistic choices.

PRISM-Bench provides a multi-dimensional, fine-grained, and human-aligned evaluation, especially a challenging "Long Text" track that leverages GCoT captions.

**Weaknesses:**

1. The paper would be far more convincing if the authors had fine-tuned an existing open-source model on this new dataset and shown corresponding performance improvements on PRISM-Bench or other benchmarks. This is the most direct way to validate the curated dataset's value.

2. The effectiveness of GCot is not evaluated. A comparison between models with and without GCoT on PRISM-Bench is encouraged.

3. An analysis comparing the proposed VLM-based evaluators (GPT-4.1, Qwen2.5-VL-72B) against traditional metrics (e.g., CLIPScore, FID) would be valuable to validate PRISM-Bench as a superior evaluation standard.

4. The abstract claims "human-aligned assessment," but the paper provides insufficient details about the human-in-the-loop refinement process. Specifically, the background, expertise and exact guidelines for the human reviewers are not described. These details are necessary to verify the quality and reliability of the claimed human alignment.

5. The paper would benefit from more qualitative visualizations of model outputs.

**Questions:**

The paper relies heavily on VLM-based automation: Qwen-VL for dataset filtering and GPT-4.1/Qwen2.5-VL-72B as judges for benchmark evaluation. However, the alignment of these automated judgments with human evaluation is not sufficiently detailed.

In dataset construction, are the filtering results of Qwen-VL aligned with human checking?

Similarly, how well do the VLM judges (GPT-4.1, Qwen2.5-VL-72B) align with human preferences and scoring on the PRISM-Bench tasks? While an automatic pipeline is valuable, its quality at each step needs to be validated against human assessment.

---

> ### Author Response · Authors · 2025-11-23
> **Response to reviewer oMcM -- Part 1**
>
> **Q1**: The paper would be far more convincing if the authors had fine-tuned an existing open-source model on this new dataset and shown corresponding performance improvements on PRISM-Bench or other benchmarks. This is the most direct way to validate the curated dataset's value. The effectiveness of GCot is not evaluated. A comparison between models with and without GCoT on PRISM-Bench is encouraged.
>
> **A1**: To demonstrate the practical utility of our dataset, we fine-tuned BAGEL [1] on FLUX-Reason-6M with two configurations: (i) using only the standard captions (no GCoT), and (ii) using an additional step where the model first predicts a reasoning text (supervised by GCoT) conditioned on the prompt, and then generates the image. Both were trained on 16 A100 GPUs for 10k steps. We evaluated the results on PRISM-Bench and GenEval [2].
> The results in Tables R1 and R2 clearly demonstrate that FLUX-Reason-6M improves model performance across multiple dimensions. Besides, GCoT further enhance model’s abilities by providing valuable structured supervision that explicitly teaches compositional reasoning, spatial understanding, and instruction following. We have added these experiments in Appendix A.2 of the revised paper.
>
> **Table R1: Performance of BAGEL fine-tuned with FLUX-Reason-6M on PRISM-Bench (part 1)**
> |    |&emsp; Imagination |  &emsp; &emsp;  Entity  |  Text rendering  | &emsp; &emsp; Style  |
> |:------:|:------:|:------:|:------:|:------:|
> |    | Ali. &emsp; Aes. &emsp; Avg. | Ali. &emsp; Aes. &emsp; Avg.  | Ali. &emsp; Aes. &emsp; Avg.  |  Ali. &emsp; Aes. &emsp; Avg.  |
> |   BAGEL(baseline) |  69.4 &emsp;	68.0 &emsp; 68.7  |    59.0 &emsp;	50.1&emsp;	54.6   |    30.2&emsp;	44.5&emsp;	37.4  |     67.9&emsp;	71.3&emsp;	69.6  |
> |  +FLUX-Reason-6M | 69.2&emsp;	75.3&emsp;	72.3 |   66.2&emsp;	58.0&emsp;	62.1   |   34.3&emsp;	44.8&emsp;	39.6  |   70.3&emsp;	81.8&emsp;	76.0  |
> |    +FLUX-Reason-6M + GCoT  |    **74.8**&emsp;	**79.5**&emsp;	**77.2** |    **71.7**&emsp;	**64.2**&emsp;	**68.0**   |     **37.4**&emsp;	**49.2**&emsp;	**43.3**  |    **75.4**	&emsp;**82.6**	&emsp;**79.0**  |
>
> **Table R1: part 2**
> |    | &emsp; &emsp; Affection  |  &emsp; Compostion |  &emsp; Long text  |  &emsp; &emsp; Overall  |
> |:------:|:------:|:------:|:------:|:------:|
> |   |  Ali. &emsp; Aes. &emsp; Avg.  |  Ali. &emsp; Aes. &emsp; Avg.  |  Ali. &emsp; Aes. &emsp; Avg.  |  Ali. &emsp; Aes. &emsp; Avg.  |
> |   BAGEL(baseline)  |    81.7&emsp;	81.4&emsp;	81.6  |    90.5&emsp;	73.1&emsp;	81.8  |    68.1&emsp;	55.3&emsp;	61.7  |    66.7&emsp;	63.4&emsp;	65.1  |
> |  +FLUX-Reason-6M |    80.9&emsp;	82.0&emsp;	81.5  |    89.4&emsp;	76.7&emsp;	83.1  |    72.3&emsp;	54.2&emsp;	63.3  |    68.9&emsp;	67.5&emsp;	68.2  |
> |    +FLUX-Reason-6M + GCoT |    **85.6**&emsp;	**90.1**&emsp;	**87.9**  |    **94.8**&emsp;	**82.7**&emsp;	**88.8**  |    **76.9**&emsp;	**61.5**&emsp;	**69.2**  |    **73.8**	&emsp;**72.8**&emsp;	**73.3**  |
>
> **Table R2: Performance of BAGEL fine-tuned with FLUX-Reason-6M on GenEval**
> |    |Single Obj.|  Two Obj. |Counting | Colors | Position |  Color Attri.  |  Overall |
> |:------:|:------:|:------:|:------:|:------:|:------:|:------:|:------:|
> |   BAGEL(baseline) | 0.99  |    0.94 | 0.81 |   0.88 |  0.64 |  0.63 |  0.82 |
> |  +FLUX-Reason-6M |**1.00** | 0.96 | 0.85|  0.89|   0.64 | 0.64 | 0.83|
> |    +FLUX-Reason-6M + GCoT  |    **1.00** |    **0.97**   |     **0.88**  |    **0.94**  |    **0.71**  |    **0.67**  |    **0.86**  |

---

> ### Author Response · Authors · 2025-11-23
> **Response to reviewer oMcM -- Part 2**
>
> **Q2**: An analysis comparing the proposed VLM-based evaluators (GPT-4.1, Qwen2.5-VL-72B) against traditional metrics (e.g., CLIPScore, FID) would be valuable to validate PRISM-Bench as a superior evaluation standard. How well do the VLM judges (GPT-4.1, Qwen2.5-VL-72B) align with human preferences and scoring on the PRISM-Bench tasks? While an automatic pipeline is valuable, its quality at each step needs to be validated against human assessment.
>
> **A2**: We have conducted a comprehensive human evaluation and compared human scores with our automatic metrics and traditional metrics.
>
> - Human evaluation protocol: We randomly sampled 20 prompts from each of the 7 tracks and generated images using 4 diverse T2I models (BAGEL, FLUX.1-dev, Qwen-Image, Gemini2.5-Flash-Image), yielding 560 image-prompt pairs. We recruited 10 well-educated graduate students to conduct human evaluation. These evaluators represented diverse disciplinary backgrounds including arts, design, and STEM fields. Each image was randomly assigned to three evaluators who conducted independent evaluations. Evaluators were instructed to rate each image according to the same prompts provided to the VLM, assessing both prompt–image alignment and aesthetic quality. They then provided a composite score (1-10) for each image based on these criteria. We averaged the three scores to obtain a human reference score for each prompt-image pair.
> - Correlation with human scores: We computed Spearman’s ρ and Kendall’s τ between human scores and three automatic evaluators, including CLIPScore [3], our VLM-based scores using Qwen2.5‑VL‑72B, and our VLM-based scores using GPT‑4.1. The results are shown in Table R3.
>
> Our VLM-based metrics achieve substantially higher correlation with human judgment than CLIPScore across all tracks. This validates PRISM-Bench as a more human-aligned evaluation standard than traditional metrics. We have included these results and detailed methodology in Section 4.3 of the revised paper.
>
> **Table R3: Correlation with Human Judgment**
> |    | Imagination| &emsp; Entity |  Text rendering | &emsp; Style |  &emsp; Affection |  Compostion |  Long text |
> |:------:|:------:|:------:|:------:|:------:|:------:|:------:|:------:|
> |    | ρ &emsp; τ |  ρ &emsp; τ  |  ρ &emsp; τ |  ρ &emsp; τ  |  ρ &emsp; τ  |  ρ &emsp; τ |  ρ &emsp; τ  |
> |   CLIPScore |  0.415&emsp;	0.306 |  0.371&emsp;	0.292   |    0.645&emsp;	0.498  |    0.427&emsp;	0.318  |   0.356&emsp;	0.283  |  0.698&emsp;	0.502  |   0.223&emsp;	0.154  |
> |  Ours(Qwen2.5-VL-72B) | 0.527&emsp;	0.382 |   0.585	&emsp;0.456 |  0.709&emsp;	0.547 |  0.658&emsp;	0.484  |    0.503&emsp;	0.372  |    0.735&emsp;	0.576  |    0.662&emsp;	0.497 |
> |    Ours(GPT-4.1)  |   **0.580**&emsp;	**0.453** |   **0.626**&emsp;	**0.481** |    **0.722**&emsp;	**0.569** |    **0.694**&emsp;	**0.531**  |   **0.559**&emsp;	**0.397**  |    **0.741**&emsp;	**0.585**|    **0.683**&emsp;	**0.510**  |
>
>
> **Q3**: The abstract claims "human-aligned assessment," but the paper provides insufficient details about the human-in-the-loop refinement process. Specifically, the background, expertise and exact guidelines for the human reviewers are not described. These details are necessary to verify the quality and reliability of the claimed human alignment.
>
> **A3**: We have added more details about the human-in-the-loop refinement in benchmark construction.
>
> - Human Reviewer Background: Five human reviewers participated in the validation of prompts during the development of PRISM-Bench: two senior PhD candidates (with 3-4 years of experience in AIGC), one master's student (with expertise in human-computer interaction research), one graphic designer (possessing over 5 years of experience in digital art and visual communication), and one postdoctoral researcher (with extensive research background in AIGC).
> - Review Guidelines: All the reviewers were instructed to evaluate each candidate prompt on
>   - Grammatical correctness and clarity
>   - Logical consistency of the described scene
>   - Feasibility for T2I generation
>   - Absence of ambiguity in spatial/attribute descriptions
>   - Alignment with the specific track's focus
>   - Absence of duplication with existing prompts
>   - Natural language flow and creative coherence
> - Consensus Protocol: Only prompts approved by all reviewers entered the final benchmark. Prompts with disagreements were revised iteratively or discarded.
>
> We have described this process and the human reviewers’ backgrounds in Section 3.1.

---

> ### Author Response · Authors · 2025-11-23
> **Response to reviewer oMcM -- Part 3**
>
> **Q4**: The paper relies heavily on VLM-based automation: Qwen-VL for dataset filtering and GPT-4.1/Qwen2.5-VL-72B as judges for benchmark evaluation. However, the alignment of these automated judgments with human evaluation is not sufficiently detailed. In dataset construction, are the filtering results of Qwen-VL aligned with human checking?
>
> **A4**: We conducted human validation of the Qwen‑VL-based filtering used in dataset construction to verify alignment between Qwen-VL's filtering results and human assessment.
>
> - We randomly sampled 500 images that Qwen‑VL had rejected and 500 that it had accepted during quality filtering.
> - 10 graduate students judged each image as “accept” or “reject” according to our criteria (blur, severe artifacts, structural failures, illegible/nonsensical text). Each image was randomly assigned to two evaluators who conducted independent evaluations.
> - An image is considered “accepted by humans” if both annotators marked it as acceptable.
>
> Finally, 95% of the images matched Qwen‑VL’s decision, indicating strong agreement between the automatic filter and human judgment. We have added this result to Appendix B.4.
>
> **Q5**: The paper would benefit from more qualitative visualizations of model outputs.
>
> **A5**: We have added more qualitative visualizations in the Appendix A.3 (Figures 7-11). Besides, we will open-source all model outputs and VLM scores for PRISM-Bench and PRISM-Bench-ZH, so that readers can inspect more examples beyond what fits in the main text.
>
> [1] Chaorui Deng, Deyao Zhu, Kunchang Li, Chenhui Gou, Feng Li, Zeyu Wang, Shu Zhong, Weihao Yu, Xiaonan Nie, Ziang Song, et al. Emerging properties in unified multimodal pretraining. arXiv preprint arXiv:2505.14683, 2025.
>
> [2] Dhruba Ghosh, Hannaneh Hajishirzi, and Ludwig Schmidt. Geneval: An object-focused framework for evaluating text-to-image alignment. Advances in Neural Information Processing Systems, 36: 52132–52152, 2023.
>
> [3] Jack Hessel, Ari Holtzman, Maxwell Forbes, Ronan Le Bras, and Yejin Choi. Clipscore: A reference-free evaluation metric for image captioning. arXiv preprint arXiv:2104.08718, 2021.

---

> > ### Comment · Reviewer_oMcM · 2025-11-25
> >
> > Thank you for the author's response. After reading, I am satisfied with the author's response. All of my concerns have been answered.
> >
> > I will raise my score.
> > Looking forward to your revised manuscript.

---

> ### Author Response · Authors · 2025-11-26
> **Thank you for acknowledging our rebuttal efforts.**
>
> Dear Reviewer oMcM,
>
> Thank you very much for your careful reading of our paper, your detailed comments, and for updating your rating after the rebuttal. The [revised manuscript](https://openreview.net/pdf?id=cPzgZnpVbN) has been updated on OpenReview. We truly appreciate your thoughtful feedback and are glad that our responses addressed your concerns.
>
> Best regards,
>
> The Authors

---

### Official Review · Reviewer_YCdw · 2025-10-30

**Soundness:** 2
**Presentation:** 3
**Contribution:** 3
**Rating:** 6
**Confidence:** 4

**Summary:**

This paper presents two contributions to the text-to-image generation field, primarily aimed at improving open-source models by focusing on complex reasoning. The contributions are: 1) FLUX-Reason-6M: A large-scale dataset of 6 million high-quality synthetic images paired with 20 million bilingual captions. The dataset is structured around six key characteristics and notably includes "Generation Chain-of-Thought" annotations to provide richer, structured supervision for reasoning. 2) PRISM-Bench: A new, 7-track benchmark designed to evaluate T2I models on these characteristics, including a "Long Text" track based on GCoT. A key feature of this benchmark is its use of advanced Vision-Language Models as judges to score both prompt-image alignment and image aesthetics, moving away from simpler, saturating metrics. The authors use this benchmark to evaluate 19 different models, identifying "Text rendering" and "Long text" comprehension as major, persistent challenges across the board. The authors intend to release both the dataset and the benchmark.

**Strengths:**

1.	The FLUX-Reason-6M dataset is impressive in its scale (6M images, 20M captions) and its use of a high-quality generator (FLUX.1-dev) ensures a strong baseline of visual quality, avoiding the noise of typical web-crawled data.

2.	The Generation Chain-of-Thought is a novel and important contribution. Providing explicit, multi-step reasoning breakdowns as a supervisory signal is a promising direction for teaching models complex compositional logic, which is a key limitation of current T2I systems.

3.	PRISM-Bench represents a significant improvement over existing T2I evaluation methods. The 7-track design allows for a much more granular analysis of model strengths and weaknesses than a single, aggregate score.

4.	The comprehensive evaluation of 19 models yields useful insights, particularly the clear confirmation that even SOTA closed-source models struggle immensely with reliable text rendering and long, complex instruction following. This helps focus the community on the next set of hard problems.

**Weaknesses:**

1.	The dataset is entirely synthetic, generated by a single model (FLUX.1-dev). This creates a significant risk of "dataset bias" or "domain gap." Models trained on FLUX-Reason-6M may learn the specific style, quirks, or common failure modes of the FLUX generator, and this improved performance might not generalize to real-world images or other domains.

2.	The PRISM-Bench relies entirely on VLM judges. While the use of VLMs is well-motivated, the central claim that these judges are "human-aligned" is asserted rather than proven. Without a correlation study comparing the VLM scores to scores from human evaluators, the validity of the benchmark is not fully established. This is a omission for a paper introducing a new evaluation standard.

3.	There is a potential self-referential loop in the methodology. The GCoT reasoning annotations are generated by a VLM, and the benchmark is then evaluated by other VLMs. This creates a system where models are trained on VLM-generated text and then evaluated by VLM judges. This VLM-in-a-loop system risks optimizing for what VLMs think is good reasoning or good alignment, which may not perfectly overlap with true human intent or logic.

**Questions:**

1.	Could the authors elaborate on the quality control process for the GCoT annotations? Specifically, how did the authors ensure that the VLM-generated reasoning steps were logical, accurate, and of high quality?

2.	The authors state that the Qwen-VL judge yields “consistent rankings” with the GPT-4.1 judge. Could the authors provide a more quantitative measure of inter-judge agreement between the two VLM judges across the seven tracks to substantiate the claim of benchmark robustness?

3.	Have the authors conducted any preliminary experiments to examine whether training on FLUX-Reason-6M, particularly with GCoT supervision, improves performance on other existing benchmarks compared with training on other large-scale datasets? If so, what do these results suggest about the practical benefits of the proposed dataset and GCoT supervision?

---

> ### Author Response · Authors · 2025-11-23
> **Response to reviewer YCdw -- Part 1**
>
> **Q1**: The dataset is entirely synthetic, generated by a single model (FLUX.1-dev). This creates a significant risk of "dataset bias" or "domain gap." Models trained on FLUX-Reason-6M may learn the specific style, quirks, or common failure modes of the FLUX generator, and this improved performance might not generalize to real-world images or other domains.
>
> **A1**: We understand your concern about potential domain gap from using FLUX.1-dev as the sole image generator. However, we'd like to highlight several design choices that mitigate this risk:
>
> - First, our extensive filtering pipeline (Section 2.3, Appendices B.2-B.3) uses Qwen2.5-VL to detect and remove typical synthesis artifacts specific to FLUX ( blur, anatomy issues, broken text). This quality control helps ensure we're curating high-fidelity samples rather than preserving model-specific quirks.
> - Second, FLUX-Reason-6M is designed as a reasoning-focused resource rather than a replacement for large-scale web datasets. The primary contribution lies in the GCoT and multi-perspective dense captions, not in reproducing natural image statistics. We envision FLUX-Reason-6M as a complementary resource to existing corpora like LAION-5B—the community can mix FLUX-Reason-6M with existing datasets to leverage GCoT for explicit reasoning supervision while maintaining visual diversity from web-scale data.
> - Third, synthetic training data is now standard practice in T2I research. We ensure semantic diversity through prompt variety (LAION-Aesthetics rewrites, imagination seed expansion, text rendering mining), so even with a fixed renderer, the semantic space (entities, relationships, emotions, styles, and so on) remains rich and diverse.
> - As shown in **Q6** below, fine-tuning a completely different architecture (BAGEL [1]) on this data improves its performance on both our PRISM-Bnech and other existing benchmark (GenEval [2]), proving that the model learns generalizable reasoning rather than memorizing FLUX artifacts.
>
> **Q2**: The PRISM-Bench relies entirely on VLM judges. While the use of VLMs is well-motivated, the central claim that these judges are "human-aligned" is asserted rather than proven. Without a correlation study comparing the VLM scores to scores from human evaluators, the validity of the benchmark is not fully established. This is a omission for a paper introducing a new evaluation standard.
>
> **A2**: We have conducted a comprehensive human evaluation and compared human scores with our automatic metrics.
>
> - Human evaluation protocol: We randomly sampled 20 prompts from each of the 7 tracks and generated images using 4 diverse T2I models (BAGEL, FLUX.1-dev, Qwen-Image, Gemini2.5-Flash-Image), yielding 560 image-prompt pairs. We recruited 10 well-educated graduate students to conduct human evaluation. These evaluators represented diverse disciplinary backgrounds including arts, design, and STEM fields. Each image was randomly assigned to three evaluators who conducted independent evaluations. Evaluators were instructed to rate each image according to the same prompts provided to the VLM, assessing both prompt–image alignment and aesthetic quality. They then provided a composite score (1-10) for each image based on these criteria. We averaged the three scores to obtain a human reference score for each prompt-image pair.
> - Correlation with human scores: We computed Spearman’s ρ and Kendall’s τ between human scores and three automatic evaluators, including CLIPScore [3], our VLM-based scores using Qwen2.5‑VL‑72B, and our VLM-based scores using GPT‑4.1. The results are shown in Table R1.
>
> Our VLM-based metrics achieve substantially higher correlation with human judgment than CLIPScore across all tracks. This validates PRISM-Bench as a more human-aligned evaluation standard than traditional metrics. We have included these results and detailed methodology in Section 4.3 of the revised paper.
>
> **Table R1: Correlation with Human Judgment**
> |    | Imagination| &emsp; Entity |  Text rendering | &emsp; Style |  &emsp; Affection |  Compostion |  Long text |
> |:------:|:------:|:------:|:------:|:------:|:------:|:------:|:------:|
> |    | ρ &emsp; τ |  ρ &emsp; τ  |  ρ &emsp; τ |  ρ &emsp; τ  |  ρ &emsp; τ  |  ρ &emsp; τ |  ρ &emsp; τ  |
> |   CLIPScore |  0.415&emsp;	0.306 |  0.371&emsp;	0.292   |    0.645&emsp;	0.498  |    0.427&emsp;	0.318  |   0.356&emsp;	0.283  |  0.698&emsp;	0.502  |   0.223&emsp;	0.154  |
> |  Ours(Qwen2.5-VL-72B) | 0.527&emsp;	0.382 |   0.585	&emsp;0.456 |  0.709&emsp;	0.547 |  0.658&emsp;	0.484  |    0.503&emsp;	0.372  |    0.735&emsp;	0.576  |    0.662&emsp;	0.497 |
> |    Ours(GPT-4.1)  |   **0.580**&emsp;	**0.453** |   **0.626**&emsp;	**0.481** |    **0.722**&emsp;	**0.569** |    **0.694**&emsp;	**0.531**  |   **0.559**&emsp;	**0.397**  |    **0.741**&emsp;	**0.585**|    **0.683**&emsp;	**0.510**  |

---

> ### Author Response · Authors · 2025-11-23
> **Response to reviewer YCdw -- Part 2**
>
> **Q3**: The authors state that the Qwen-VL judge yields “consistent rankings” with the GPT-4.1 judge. Could the authors provide a more quantitative measure of inter-judge agreement between the two VLM judges across the seven tracks to substantiate the claim of benchmark robustness?
>
> **A3**: We computed inter-judge correlation (Spearman’s ρ and Kendall’s τ) between GPT-4.1 and Qwen2.5-VL-72B across the average scores of all 19 models on each track. The results are in Table R2.
> The consistently high correlation (ρ > 0.86 across all tracks, average 0.925) demonstrates remarkable agreement between the two judges despite their different architectures and training. The near-perfect agreement on Text Rendering (ρ=0.978) and Long Text (ρ=0.982) is especially encouraging for our most challenging tracks. This validates the robustness and reliability of PRISM-Bench's evaluation protocol. We have added the inter-judge agreement analysis to Section 4.3 of the revised manuscript.
>
> **Table R2: Inter-Judge Agreement (GPT-4.1 vs. Qwen2.5-VL)**
> |    | Imagination| &emsp; Entity |  Text rendering | &emsp; Style |  &emsp; Affection |  Compostion |  Long text |
> |:------:|:------:|:------:|:------:|:------:|:------:|:------:|:------:|
> |    | ρ &emsp; τ |  ρ &emsp; τ  |  ρ &emsp; τ |  ρ &emsp; τ  |  ρ &emsp; τ  |  ρ &emsp; τ |  ρ &emsp; τ  |
> |  GPT & Qwen-VL|  0.861&emsp;	0.720 |  0.872&emsp;	0.704  |   0.978&emsp;	0.903 |    0.973&emsp;	0.888  |  0.872&emsp;	0.739 |  0.935&emsp;	0.806|   0.982&emsp;	0.947  |
>
> **Q4**: There is a potential self-referential loop in the methodology. The GCoT reasoning annotations are generated by a VLM, and the benchmark is then evaluated by other VLMs. This creates a system where models are trained on VLM-generated text and then evaluated by VLM judges. This VLM-in-a-loop system risks optimizing for what VLMs think is good reasoning or good alignment, which may not perfectly overlap with true human intent or logic.
>
> **A4**:  Thank you for this question. We'd like to respectfully clarify a potential misunderstanding about our methodology. GCoT annotations in FLUX-Reason-6M are generated by VLM. However, what we evaluate in PRISM-Bench are images generated by external T2I models, which is unrelated to GCoT. The T2I models we evaluate (GPT-Image-1, Qwen-Image, Stable Diffusion 3.5, etc.) are pretrained models that have never seen our GCoT annotations. Our benchmark serves purely to measure how well existing T2I models handle multi-dimensional complex prompts. Thus, there is no circular optimization.
>
> **Q5**: Could the authors elaborate on the quality control process for the GCoT annotations? Specifically, how did the authors ensure that the VLM-generated reasoning steps were logical, accurate, and of high quality?
>
> **A5**: Our GCoT quality control involves multiple stages.
>
> - Input filtering: Only images that pass our strict VLM quality checks (Section 2.3) and receive high scores on relevant characteristics receive GCoT annotations. This ensures the visual foundation is reliable.
> - Prompt engineering: We designed detailed prompts for GCoT generation that require the powerful VLM to integrate all six category-specific captions (when applicable), maintain consistency with factual captions (entity names, text content, spatial relations), and produce coherent step-by-step narratives about scene composition.
> - Human validation: During the rebuttal period, we randomly sampled 500 images with their GCoT annotations and asked 10 graduate students to identify any errors, including hallucinations, contradictions with factual captions, or logical inconsistencies. 96% of GCoTs passed this review, demonstrating high annotation quality. The 4% failure cases typically involved minor attribute mismatches rather than fundamental logical errors.
>
> We have added these quality control details to Appendix B.5 of the revised manuscript.

---

> ### Author Response · Authors · 2025-11-23
> **Response to reviewer YCdw -- Part 3**
>
> **Q6**: Have the authors conducted any preliminary experiments to examine whether training on FLUX-Reason-6M, particularly with GCoT supervision, improves performance on other existing benchmarks compared with training on other large-scale datasets? If so, what do these results suggest about the practical benefits of the proposed dataset and GCoT supervision?
>
> **A6**: To demonstrate the practical utility of our dataset, we fine-tuned BAGEL [1] on FLUX-Reason-6M with two configurations: (i) using only the standard captions (no GCoT), and (ii) using an additional step where the model first predicts a reasoning text (supervised by GCoT) conditioned on the prompt, and then generates the image. Both were trained on 16 A100 GPUs for 10k steps. We evaluated the results on PRISM-Bench and GenEval [2].
> The results in Tables R3 and R4 indicate that FLUX‑Reason‑6M does deliver practical benefits and that GCoT supervision further strengthens compositional and instruction-following abilities, even on an external benchmark. This suggests that the reasoning skills learned from our dataset are not confined to our own benchmark but are generalizable. We have included these results in Appendix A.2 of the revised paper.
>
> **Table R3: Performance of BAGEL fine-tuned with FLUX-Reason-6M on PRISM-Bench (part 1)**
> |    |&emsp; Imagination |  &emsp; &emsp;  Entity  |  Text rendering  | &emsp; &emsp; Style  |
> |:------:|:------:|:------:|:------:|:------:|
> |    | Ali. &emsp; Aes. &emsp; Avg. | Ali. &emsp; Aes. &emsp; Avg.  | Ali. &emsp; Aes. &emsp; Avg.  |  Ali. &emsp; Aes. &emsp; Avg.  |
> |   BAGEL(baseline) |  69.4 &emsp;	68.0 &emsp; 68.7  |    59.0 &emsp;	50.1&emsp;	54.6   |    30.2&emsp;	44.5&emsp;	37.4  |     67.9&emsp;	71.3&emsp;	69.6  |
> |  +FLUX-Reason-6M | 69.2&emsp;	75.3&emsp;	72.3 |   66.2&emsp;	58.0&emsp;	62.1   |   34.3&emsp;	44.8&emsp;	39.6  |   70.3&emsp;	81.8&emsp;	76.0  |
> |    +FLUX-Reason-6M + GCoT  |    **74.8**&emsp;	**79.5**&emsp;	**77.2** |    **71.7**&emsp;	**64.2**&emsp;	**68.0**   |     **37.4**&emsp;	**49.2**&emsp;	**43.3**  |    **75.4**	&emsp;**82.6**	&emsp;**79.0**  |
>
> **Table R3: part 2**
> |    | &emsp; &emsp; Affection  |  &emsp; Compostion |  &emsp; Long text  |  &emsp; &emsp; Overall  |
> |:------:|:------:|:------:|:------:|:------:|
> |   |  Ali. &emsp; Aes. &emsp; Avg.  |  Ali. &emsp; Aes. &emsp; Avg.  |  Ali. &emsp; Aes. &emsp; Avg.  |  Ali. &emsp; Aes. &emsp; Avg.  |
> |   BAGEL(baseline)  |    81.7&emsp;	81.4&emsp;	81.6  |    90.5&emsp;	73.1&emsp;	81.8  |    68.1&emsp;	55.3&emsp;	61.7  |    66.7&emsp;	63.4&emsp;	65.1  |
> |  +FLUX-Reason-6M |    80.9&emsp;	82.0&emsp;	81.5  |    89.4&emsp;	76.7&emsp;	83.1  |    72.3&emsp;	54.2&emsp;	63.3  |    68.9&emsp;	67.5&emsp;	68.2  |
> |    +FLUX-Reason-6M + GCoT |    **85.6**&emsp;	**90.1**&emsp;	**87.9**  |    **94.8**&emsp;	**82.7**&emsp;	**88.8**  |    **76.9**&emsp;	**61.5**&emsp;	**69.2**  |    **73.8**	&emsp;**72.8**&emsp;	**73.3**  |
>
> **Table R4: Performance of BAGEL fine-tuned with FLUX-Reason-6M on GenEval**
> |    |Single Obj.|  Two Obj. |Counting | Colors | Position |  Color Attri.  |  Overall |
> |:------:|:------:|:------:|:------:|:------:|:------:|:------:|:------:|
> |   BAGEL(baseline) | 0.99  |    0.94 | 0.81 |   0.88 |  0.64 |  0.63 |  0.82 |
> |  +FLUX-Reason-6M |**1.00**| 0.96 | 0.85|  0.89|   0.64 | 0.64 | 0.83|
> |    +FLUX-Reason-6M + GCoT  |    **1.00** |    **0.97**   |     **0.88**  |    **0.94**  |    **0.71**  |    **0.67**  |    **0.86**  |
>
> [1] Chaorui Deng, Deyao Zhu, Kunchang Li, Chenhui Gou, Feng Li, Zeyu Wang, Shu Zhong, Weihao Yu, Xiaonan Nie, Ziang Song, et al. Emerging properties in unified multimodal pretraining. arXiv preprint arXiv:2505.14683, 2025.
>
> [2] Dhruba Ghosh, Hannaneh Hajishirzi, and Ludwig Schmidt. Geneval: An object-focused framework for evaluating text-to-image alignment. Advances in Neural Information Processing Systems, 36: 52132–52152, 2023.
>
> [3] Jack Hessel, Ari Holtzman, Maxwell Forbes, Ronan Le Bras, and Yejin Choi. Clipscore: A reference-free evaluation metric for image captioning. arXiv preprint arXiv:2104.08718, 2021.

---

### Official Review · Reviewer_rbSW · 2025-11-04

**Soundness:** 3
**Presentation:** 3
**Contribution:** 4
**Rating:** 6
**Confidence:** 4

**Summary:**

This work present a training dataset, FLUX-Reason-6M, and a benchmark, PRISM-Bench. FLUX-Reason-6M contains 6M FLUX-generated images paired with carefully curated prompts that requires reasoning during image generation. Each text-image pair comes with multiple captions and a Generation Chain-of-Thought (GCoT) that provide details breakdowns of the image generation process.

The PRISM-Bench is a challenging benchmark that covers the complex reasoning abilities introduced in FLUX-Reason-5M. The benchmark uses strong MLLMs like GPT-4.1 and Qwen2.5-VL-72B to do evaluations. Experiments show big performance gaps across models, and hights the areas that need improvements.

**Strengths:**

1. The field needs resources toward more complex and reasoning-heavy text-to-image generation tasks. This paper provides both the training data and the evaluation benchmark for it.

2. The paper contains detailed description of the carefully designed data curation pipeline. The pipeline make sense and can generate data of high quality. The benchmark is further examined by humans to ensure quality. These make the datasets included in this paper valuable to the community.

3. The paper is well-written and easy to read.

**Weaknesses:**

1. The dataset FLUX-Reason-6M only used a single text-to-image model, FLUX.1-dev to create images. It potentially has the model's biases and inherits its failure modes.

2. There is no experiments that trains any model with FLUX-Reason-6M. There lacks evidence that this dataset is helpful for model training.

3. PRISM-Bench relies on MLLM for evaluation. However, MLLMs can be unreliable, especially with these challenging tasks. There is no human experiments that show how well this benchmark aligns with humans.

**Questions:**

1. Have you tried train any model with FLUX-Reason-6M? Can it yield improvements on tasks?

2. Is there any evidence that GCOT can improve model training? Any ablation experiments showing that?

3. Have you conduct any human evaluation on PRISM-Bench? How well does the evaluation metrics align with human judgements?

---

> ### Author Response · Authors · 2025-11-23
> **Response to reviewer rbSW -- Part 1**
>
> **Q1**: The dataset FLUX-Reason-6M only used a single text-to-image model, FLUX.1-dev to create images. It potentially has the model's biases and inherits its failure modes.
>
> **A1**: We acknowledge that relying on a single generator risks inheriting specific biases. However, we have mitigated this through several measures, and our new experiments confirm the data's transferability:
>
> - First, as detailed in Section 2.3 and Appendix B, we utilize Qwen2.5-VL to aggressively filter out images containing "FLUX-specific" failure modes (artifacts, text distortions, etc.). This ensures the dataset represents high-quality visual concepts rather than model-specific noise.
> - Second, FLUX-Reason-6M is designed as a reasoning-focused resource rather than a replacement for large-scale web datasets. The primary contribution lies in the GCoT and multi-perspective captions, not in reproducing natural image statistics. We envision FLUX-Reason-6M as a complementary resource to existing corpora like LAION-5B—the community can mix FLUX-Reason-6M with existing datasets to utilize GCoT for explicit reasoning supervision while maintaining visual diversity from web-scale data.
> - Third, synthetic training data is now standard practice in T2I research. We ensure semantic diversity through prompt variety (LAION-Aesthetics rewrites, imagination seed expansion, text rendering mining), so even with a fixed renderer, the semantic space (entities, relationships, emotions, styles, etc.) remains diverse.
> - As shown in **Q2** below, fine-tuning a different architecture (BAGEL [1]) on this data improves its performance, proving that the model learns generalizable reasoning rather than memorizing FLUX artifacts.
>
> **Q2**: There is no experiments that trains any model with FLUX-Reason-6M. There lacks evidence that this dataset is helpful for model training. Is there any evidence that GCOT can improve model training?
>
> **A2**: To demonstrate the practical utility of our dataset, we fine-tuned BAGEL [1] on FLUX-Reason-6M with two configurations: (i) using only the standard captions (no GCoT), and (ii) using an additional step where the model first predicts a reasoning text (supervised by GCoT) conditioned on the prompt, and then generates the image. Both were trained on 16 A100 GPUs for 10k steps. We evaluated the results on PRISM-Bench and GenEval [2].
> The results in Tables R1 and R2 clearly demonstrate that FLUX-Reason-6M improves model performance across multiple dimensions. Besides, GCoT further enhance model’s abilities by providing valuable structured supervision that explicitly teaches compositional reasoning, spatial understanding, and instruction following. We have added these experiments in Appendix A.2 of the revised paper.
>
> **Table R1: Performance of BAGEL fine-tuned with FLUX-Reason-6M on PRISM-Bench (part 1)**
> |    |&emsp; Imagination |  &emsp; &emsp;  Entity  |  Text rendering  | &emsp; &emsp; Style  |
> |:------:|:------:|:------:|:------:|:------:|
> |    | Ali. &emsp; Aes. &emsp; Avg. | Ali. &emsp; Aes. &emsp; Avg.  | Ali. &emsp; Aes. &emsp; Avg.  |  Ali. &emsp; Aes. &emsp; Avg.  |
> |   BAGEL(baseline) |  69.4 &emsp;	68.0 &emsp; 68.7  |    59.0 &emsp;	50.1&emsp;	54.6   |    30.2&emsp;	44.5&emsp;	37.4  |     67.9&emsp;	71.3&emsp;	69.6  |
> |  +FLUX-Reason-6M | 69.2&emsp;	75.3&emsp;	72.3 |   66.2&emsp;	58.0&emsp;	62.1   |   34.3&emsp;	44.8&emsp;	39.6  |   70.3&emsp;	81.8&emsp;	76.0  |
> |    +FLUX-Reason-6M + GCoT  |    **74.8**&emsp;	**79.5**&emsp;	**77.2** |    **71.7**&emsp;	**64.2**&emsp;	**68.0**   |     **37.4**&emsp;	**49.2**&emsp;	**43.3**  |    **75.4**	&emsp;**82.6**	&emsp;**79.0**  |
>
> **Table R1: part 2**
> |    | &emsp; &emsp; Affection  |  &emsp; Compostion |  &emsp; Long text  |  &emsp; &emsp; Overall  |
> |:------:|:------:|:------:|:------:|:------:|
> |   |  Ali. &emsp; Aes. &emsp; Avg.  |  Ali. &emsp; Aes. &emsp; Avg.  |  Ali. &emsp; Aes. &emsp; Avg.  |  Ali. &emsp; Aes. &emsp; Avg.  |
> |   BAGEL(baseline)  |    81.7&emsp;	81.4&emsp;	81.6  |    90.5&emsp;	73.1&emsp;	81.8  |    68.1&emsp;	55.3&emsp;	61.7  |    66.7&emsp;	63.4&emsp;	65.1  |
> |  +FLUX-Reason-6M |    80.9&emsp;	82.0&emsp;	81.5  |    89.4&emsp;	76.7&emsp;	83.1  |    72.3&emsp;	54.2&emsp;	63.3  |    68.9&emsp;	67.5&emsp;	68.2  |
> |    +FLUX-Reason-6M + GCoT |    **85.6**&emsp;	**90.1**&emsp;	**87.9**  |    **94.8**&emsp;	**82.7**&emsp;	**88.8**  |    **76.9**&emsp;	**61.5**&emsp;	**69.2**  |    **73.8**	&emsp;**72.8**&emsp;	**73.3**  |
>
> **Table R2: Performance of BAGEL fine-tuned with FLUX-Reason-6M on GenEval**
> |    |Single Obj.|  Two Obj. |Counting | Colors | Position |  Color Attri.  |  Overall |
> |:------:|:------:|:------:|:------:|:------:|:------:|:------:|:------:|
> |   BAGEL(baseline) | 0.99  |    0.94 | 0.81 |   0.88 |  0.64 |  0.63 |  0.82 |
> |  +FLUX-Reason-6M |**1.00** | 0.96 | 0.85|  0.89|   0.64 | 0.64 | 0.83|
> |    +FLUX-Reason-6M + GCoT  |    **1.00** |    **0.97**   |     **0.88**  |    **0.94**  |    **0.71**  |    **0.67**  |    **0.86**  |

---

> ### Author Response · Authors · 2025-11-23
> **Response to reviewer rbSW -- Part 2**
>
> **Q3**: PRISM-Bench relies on MLLM for evaluation. However, MLLMs can be unreliable, especially with these challenging tasks. There is no human experiments that show how well this benchmark aligns with humans.
>
> **A3**: We have conducted a comprehensive human evaluation and compared human scores with our automatic metrics.
>
> - Human evaluation protocol: We randomly sampled 20 prompts from each of the 7 tracks and generated images using 4 diverse T2I models (BAGEL, FLUX.1-dev, Qwen-Image, Gemini2.5-Flash-Image), yielding 560 image-prompt pairs. We recruited 10 well-educated graduate students to conduct human evaluation. These evaluators represented diverse disciplinary backgrounds including arts, design, and STEM fields. Each image was randomly assigned to three evaluators who conducted independent evaluations. Evaluators were instructed to rate each image according to the same prompts provided to the VLM, assessing both prompt–image alignment and aesthetic quality. They then provided a composite score (1-10) for each image based on these criteria. We averaged the three scores to obtain a human reference score for each prompt-image pair.
> - Correlation with human scores: We computed Spearman’s ρ and Kendall’s τ between human scores and three automatic evaluators, including CLIPScore [3], our VLM-based scores using Qwen2.5‑VL‑72B, and our VLM-based scores using GPT‑4.1. The results are shown in Table R3.
>
> Our VLM-based metrics achieve substantially higher correlation with human judgment than CLIPScore across all tracks. This validates PRISM-Bench as a more human-aligned evaluation standard than traditional metrics. We have included these results and detailed methodology in Section 4.3 of the revised paper.
>
> **Table R3: Correlation with Human Judgment**
> |    | Imagination| &emsp; Entity |  Text rendering | &emsp; Style |  &emsp; Affection |  Compostion |  Long text |
> |:------:|:------:|:------:|:------:|:------:|:------:|:------:|:------:|
> |    | ρ &emsp; τ |  ρ &emsp; τ  |  ρ &emsp; τ |  ρ &emsp; τ  |  ρ &emsp; τ  |  ρ &emsp; τ |  ρ &emsp; τ  |
> |   CLIPScore |  0.415&emsp;	0.306 |  0.371&emsp;	0.292   |    0.645&emsp;	0.498  |    0.427&emsp;	0.318  |   0.356&emsp;	0.283  |  0.698&emsp;	0.502  |   0.223&emsp;	0.154  |
> |  Ours(Qwen2.5-VL-72B) | 0.527&emsp;	0.382 |   0.585	&emsp;0.456 |  0.709&emsp;	0.547 |  0.658&emsp;	0.484  |    0.503&emsp;	0.372  |    0.735&emsp;	0.576  |    0.662&emsp;	0.497 |
> |    Ours(GPT-4.1)  |   **0.580**&emsp;	**0.453** |   **0.626**&emsp;	**0.481** |    **0.722**&emsp;	**0.569** |    **0.694**&emsp;	**0.531**  |   **0.559**&emsp;	**0.397**  |    **0.741**&emsp;	**0.585**|    **0.683**&emsp;	**0.510**  |
>
> [1] Chaorui Deng, Deyao Zhu, Kunchang Li, Chenhui Gou, Feng Li, Zeyu Wang, Shu Zhong, Weihao Yu, Xiaonan Nie, Ziang Song, et al. Emerging properties in unified multimodal pretraining. arXiv preprint arXiv:2505.14683, 2025.
>
> [2] Dhruba Ghosh, Hannaneh Hajishirzi, and Ludwig Schmidt. Geneval: An object-focused framework for evaluating text-to-image alignment. Advances in Neural Information Processing Systems, 36: 52132–52152, 2023.
>
> [3] Jack Hessel, Ari Holtzman, Maxwell Forbes, Ronan Le Bras, and Yejin Choi. Clipscore: A reference-free evaluation metric for image captioning. arXiv preprint arXiv:2104.08718, 2021.

---

### Author Response · Authors · 2025-12-02
**Review and Reviewer-Author Discussion Summary [1/2]**

Dear PCs, SACs, ACs, and Reviewers,

Thank you very much for your valuable contributions to our work. To assist the newly assigned AC and help reduce their workload, we provide below a summary of the key points from the reviews and the reviewer-author discussions.

---

**Strengths.**  Overall, we are grateful that the reviewers gave this paper a positive evaluation in the initial reviews, with all three reviewers rating it above the acceptance threshold in the discussion period. Specifically:

- **A large-scale, reasoning-focused T2I dataset (FLUX-Reason-6M).**

  All three reviewers recognized this contribution (rbSW: Strength 1, YCdw: Strength 1, oMcM: Strength 1).

- **Novel Generation Chain-of-Thought (GCoT) as structured supervision.**

  All three reviewers highlighted this innovation (rbSW: mentioned in summary, YCdw: Strength 2, oMcM: Strength 2).

- **PRISM-Bench as a comprehensive, multi-track benchmark.**

    All three reviewers appreciated this contribution (rbSW: mentioned in summary, YCdw: Strength 3, oMcM: Strength 3).

- **Insightful large-scale evaluation of 19 models.**

    All three reviewers acknowledged this (rbSW: mentioned in summary, YCdw: Strength 4, oMcM: mentioned in summary).

- **Quality of curation pipeline, human involvement, and writing.**

    Two reviewers recognized this strength (rbSW: Strength 2-3, YCdw: acknowledged in summary).

---

> ### Author Response · Authors · 2025-12-02
> **Review and Reviewer-Author Discussion Summary [2/2]**
>
> **Concerns and Our Addressing.** During the discussion period, we actively addressed the reviewers' concerns through additional experiments and clarifications, which were recognized by the reviewers and led to increased scores. Specifically:
>
> - **Concerns about dataset validity and generalization**
>
>     - (rbSW: Question 1-2, YCdw: Question 3, oMcM: Weakness 1-2) **Lack of training experiments to demonstrate dataset utility and GCoT effectiveness.**
>
>         **Our Addressing.** We conducted comprehensive fine-tuning experiments with BAGEL in three configurations: (a) baseline, (b) +FLUX-Reason-6M, (c) +FLUX-Reason-6M + GCoT. Results in **Tables R1-R2** in our response to reviewer rbSW show clear improvements across all dimensions, with GCoT providing additional gains. Reviewer oMcM explicitly acknowledged this addresses the concerns.
>     - (rbSW: Weakness 1, YCdw: Weakness 1) **Using a single model for image generation may introduce biases.**
>
>         **Our Addressing.** We clarified our mitigation strategies: (a) aggressive VLM-based filtering removes FLUX-specific artifacts, (b) the dataset serves as a complementary reasoning-focused resource rather than a replacement for web-scale data, (c) semantic diversity is ensured through varied prompt sources. Crucially, we added fine-tuning experiments showing that a different architecture (BAGEL) improves performance on both PRISM-Bench and GenEval, proving the data teaches generalizable reasoning rather than model-specific patterns. Please refer our responses to reviewers rbSW and YCdw for details.
>
> - **Concerns about alignment between evaluation results and humans**
>     - (rbSW: Weakness 3, YCdw: Weakness 2, oMcM: Weakness 3, Questions 3) **VLM-based evaluation lacks validation against human judgment.**
>
>         **Our Addressing.** We conducted a comprehensive human evaluation study: 10 graduate students from diverse backgrounds (arts, design, STEM) evaluated 560 image-prompt pairs. Results in **Table R3** in our response to reviewer rbSW show our VLM-based metrics achieve substantially higher correlation with human judgment than CLIPScore across all tracks. Reviewer oMcM explicitly acknowledged this addresses the concerns.
>
>     - (YCdw: Question 2) **Inter-judge reliability between GPT-4.1 and Qwen2.5-VL needs quantitative validation.**
>
>         **Our Addressing.** We added inter-judge agreement analysis in in **Table R2** of our response to reviewer YCdw showing consistently high correlation, demonstrating robust and reliable evaluation.
>
> - **Concerns about quality control and methodology**
>     - (YCdw: Question 1, oMcM: Question 2) **Quality control for GCoT annotations and VLM filtering needs validation.**
>
>         **Our Addressing.** For GCoT quality: We added detailed quality control process (see **A5** in our response to reviewer YCdw) and conducted human validation of 500 randomly sampled GCoT annotations by 10 graduate students, achieving 96% pass rate. For VLM filtering: We validated Qwen-VL filtering decisions against human judgment on 1,000 images (500 accepted + 500 rejected), achieving 95% agreement (see **A4** in our response to reviewer oMcM). Reviewer oMcM acknowledged these validations.
>
>     - (YCdw: Weakness 3) **Potential self-referential loop: VLM generates GCoT, VLM judges benchmark.**
>
>         **Our Addressing.** We clarified that there is no circular optimization: the T2I models evaluated on PRISM-Bench (GPT-Image-1, Qwen-Image, SD3.5, etc.) are pretrained models that have never seen our GCoT annotations. The benchmark purely measures how well existing models handle complex prompts. Please refer to **A4** in our response to reviewer YCdw.
>
>     - (oMcM: Weakness 4) **Insufficient details about human-in-the-loop refinement process for benchmark prompts.**
>
>         **Our Addressing.** We added comprehensive details: (a) reviewer backgrounds (2 senior PhDs in AIGC, 1 HCI master's student, 1 graphic designer with 5+ years experience, 1 postdoc in AIGC), (b) detailed evaluation criteria (7 specific dimensions), (c) consensus protocol (unanimous approval required). Reviewer oMcM acknowledged this addresses the concern.
>
> - **Presentation improvements**
>     - (oMcM: Weakness 5) **Need more qualitative visualizations.**
>
>         **Our Addressing.** We added extensive visualizations in [Revision](https://openreview.net/pdf?id=cPzgZnpVbN) (Appendix A.3, Figures 7-11) covering all benchmark tracks. Reviewer oMcM acknowledged this improvement.
>
> **Recognition of our revision from reviewers.** One reviewer (oMcM) explicitly stated "All of my concerns have been answered" and raised the score from 4 to 6. We believe that we have also properly addressed the remaining two reviewers' concerns.
>
> ---
>
> We hope this concise summary will assist the AC's work. Thank you again to all reviewers and chairs for their constructive feedback, which has significantly strengthened our paper.
>
> Sincerely,
>
> The Authors

---

### Meta-Review · Area_Chair_igVH · 2026-01-14

**Summary:**

This paper introduces FLUX-Reason-6M for complex reasoning and PRISM-Bench as a comprehensive text-to-image benchmark. The major concerns are: 1. effectiveness of dataset and analysis of dataset bias. 2. quality controls for GCoT annotations and VLM filtering. 3. the alignment between VLM-based evaluation and human judgment. The authors address these concerns comprehensively with convincing experiment results. Therefore, I recommend accepting this paper.

**Reviewer Concerns:**

The major concerns including the effectiveness of dataset, quality controls for GCoT annotations and VLM filtering, and the alignment between VLM-based evaluation and human judgment habe been well addressed by the rebuttal. Regarding the bias introduced by the FLUX model, the authors state that FLUX-Reason-6M is designed as a reasoning-focused resource rather than a replacement for large-scale web datasets. While the response is reasonable, it would make this work even better if more models can be used to mitigate the model bias.

**Reviewer Scores:**

Overall, most of the concerns have been addressed by the rebuttal. Some reviewers raised their scores after the rebuttal.

---

### Decision · Program_Chairs · 2026-01-26

Accept (Poster)